# Consensus model of a cyanobacterial light-dependent protochlorophyllide oxidoreductase in its pigment-free apo-form and photoactive ternary complex

Judith Schneidewind[1], Frank Krause[2], Marco Bocola[3], Andreas Maximilian Stadler [4,5], Mehdi D. Davari[3], Ulrich Schwaneberg[3,6], Karl-Erich Jaeger [1,7] & Ulrich Krauss [1,7]

Photosynthetic organisms employ two different enzymes for the reduction of the C17 = C18 double bond of protochlorophyllide (Pchlide), yielding the chlorophyll precursor chlorophyllide. First, a nitrogenase-like, light-independent (dark-operative) Pchlide oxidoreductase and secondly, a light-dependent Pchlide oxidoreductase (LPOR). For the latter enzyme, despite decades of research, no structural information is available. Here, we use protein structure modelling, molecular dynamics (MD) simulations combined with multi-wavelength analytical ultracentrifugation (MWA-AUC) and small angle X-ray scattering (SAXS) experiments to derive a consensus model of the LPOR apoprotein and the substrate/cofactor/LPOR ternary complex. MWA-AUC and SAXS experiments independently demonstrate that the apoprotein is monomeric, while ternary complex formation induces dimerization. SAXS-guided modelling studies provide a full-length model of the apoprotein and suggest a tentative mode of dimerization for the LPOR ternary complex, supported by published cross-link constraints. Our study provides a first impression of the LPOR structural organization.

[1] Institut für Molekulare Enzymtechnologie, Heinrich-Heine-Universität Düsseldorf, Forschungszentrum Jülich GmbH, D-52425 Jülich, Germany. [2] Nanolytics, Gesellschaft für Kolloidanalytik GmbH, Am Mühlenberg 11, 14476 Potsdam, Germany. [3] Lehrstuhl für Biotechnologie, RWTH Aachen University, Worringerweg 3, 52074 Aachen, Germany. [4] Jülich Centre for Neutron Science (JCNS-1) and Institute for Complex Systems (ICS-1), Forschungszentrum Jülich GmbH, D-52425 Jülich, Germany. [5] Institute of Physical Chemistry, RWTH Aachen University, Landoltweg 2, 52056 Aachen, Germany. [6] DWI-Leibniz Institut für Interaktive Materialien, Forckenbeckstraße 50, 52056 Aachen, Germany. [7] IBG-1: Biotechnologie, Forschungszentrum Jülich GmbH, D-52425 Jülich, Germany. Correspondence and requests for materials should be addressed to U.K. (email: u.krauss@fz-juelich.de)

Light-dependent protochlorophyllide oxidoreductases (LPOR, E.C. 1.3.1.33) are photoenzymes that catalyze the strictly light-dependent reduction of the C17 = C18 double bond of protochlorophyllide (Pchlide) to yield chlorophyllide (Chlide) using NADPH as electron donor (Fig. 1a)[1–4]. LPORs are ideal model systems to study biological hydride transfer reactions because the photoactive LPOR/NADPH/Pchlide holoprotein ternary complex (hereafter "holoprotein") can be reconstituted in the dark with catalysis initiated by short pulses of light[1,2,4–6]. Ultrafast and cryogenic spectroscopy have revealed the reaction mechanism of LPORs in great detail[6–12], yet structural information for this important enzyme family is lacking. Homology models have been based on short-chain dehydrogenases with a conserved Rossmann-fold[1,10,13–17] but the quaternary structure of the enzyme remains unresolved with conflicting information for the NADPH/Pchlide-free apoprotein ("apoprotein") and the corresponding holoprotein complexes. A pea LPOR / maltose-binding protein (MBP) fusion protein eluted as a dimer in size-exclusion chromatography (SEC) irrespective of substrate and product being present[18]. *Arabidopsis thaliana* LPOR similarly forms dimers[19], whereas *in vitro* cross-linking of recombinant LPOR A of *A. thaliana* (*At*LPOR A) revealed that both apoprotein and holoprotein form tightly packed, higher-order oligomers[20]. Homology modelling and non-denaturing protein blots indicated that barley LPOR forms hexamers with five LPOR A and one LPOR B subunit[19]. Whereas early LPOR preparations were generally derived from solubilized etioplasts[21,22], recently plant and cyanobacterial LPORs have been recombinantly produced in *Escherichia coli*[6,9,11,13,20,23]. Plant LPORs tend to aggregate limiting structural and biophysical analyses. Cyanobacterial LPORs, e.g. from *Thermosynechococcus elongatus* BP-1 (*Te*LPOR) or *Synechocystis* sp. (*Ss*LPOR), by contrast, can be produced in sufficient quantity and purity for spectroscopic/ biophysical characterization. Recent information on the LPOR reaction mechanism thus primarily derives from cyanobacterial LPORs[2,5–7,9,10,24]. Both the structure and oligomerization state of cyanobacterial LPORs, however, remain unknown.

To fill this gap, we investigated the structure of cyanobacterial *Te*LPOR apoprotein in solution and LPOR complex formation by supplementing NADPH and Pchlide. Multi-wavelength analytical ultracentrifugation (MWA-AUC), small angle X-ray scattering (SAXS) and complementary biochemical and biophysical analyses indicate apo-*Te*LPOR to be monomeric. SAXS-derived *ab initio* models reveal a bowling-pin like molecule in line with existing LPOR homology models, while adding a C-terminal extension not previously proposed. SAXS and MWA-AUC studies of the holoprotein complex show that substrate/cofactor binding results in dimerization of *Te*LPOR. Tentative protein-docking models of the *Te*LPOR dimer corroborated by MD simulations indicate potential dimerization via the active site surface, conserved structural elements on the SDR-like Rossmann fold and protruding C-terminal α-helices. This dimerization mode corroborates our SAXS data and recent cross-linking data for a plant LPOR. Our study thus sheds light on the structural organization of this important class of enzymes.

## Results

### Holoprotein assembly, functionality and oligomerization.
The N-terminally His$_6$-tagged *Te*LPOR apoprotein was heterologously produced in *E. coli*, purified to homogeneity by immobilized metal ion affinity chromatography (IMAC) and preparative size exclusion chromatography (SEC) (Supplementary Fig. 1a, b). The *Te*LPOR holoprotein, was reconstituted by mixing equimolar amounts of apoprotein, Pchlide and NADPH. To verify holoprotein functionality, samples were diluted in reaction buffer and

illuminated with blue light for different time intervals. Corresponding light-dark difference spectra (Fig. 1b) reveal enzyme-bound Pchlide decreasing (peak at 642 nm)[8] and concomitant Chlide product (new peak at 675 nm)[12] formation. For details regarding *Te*LPOR purification, reconstitution and functionality tests refer to the Supplementary Information (Supplementary Discussion Section 1.1, Supplementary Fig. 1, Supplementary Fig. 2).

To evaluate sample monodispersity, purity and test for changes in the oligomerization state during holoprotein reconstitution, we analysed an apoprotein sample by SEC and SDS–PAGE (Supplementary Fig. 1b, c) and performed a series of multi-wavelength absorbance analytical ultra-centrifugation (MWA-AUC) experiments for the apo and holoprotein (Supplementary Fig. 3, Fig. 1c–e) at concentrations from 0.5 to 1.23 mg ml$^{-1}$. The sedimentation coefficient distribution $c(s)$ of the 0.615 mg ml$^{-1}$ apoprotein sample indicates that monomers of 43 kDa (1.01 S) constitute 94.2% of the sample, while dimers and larger oligomers/aggregates constitute 3.6 and 2.2%, respectively (Fig. 1c and Supplementary Fig. 3a, c) - in excellent agreement with the SAXS analyses for similar apoprotein samples below (Supplementary Tables 1 and 2).

While AUC analyses of the apoprotein are straightforward, AUC analysis of the holoprotein is complicated by the presence of low concentrations of free NADPH and Pchlide (due to the low µM $K_d$ of the ternary holoprotein complex[10]), spectral shifts and altered extinction coefficients associated with Pchlide binding[5,25]. In contrast to the largely monomeric apoprotein, the $c(s)$ distribution of a ~0.5 mg ml$^{-1}$ holoprotein sample at 279, 340 and 437 nm indicate larger amounts of more rapidly sedimenting species such as protein dimers and various non-protein aggregates (Fig. 1d, Supplementary Fig. 3c). Free NADPH and Pchlide that sediment more rapidly than *Te*LPOR may be seen in the propagation of typical absorption maxima of moving sedimentation boundaries (Supplementary Movie 1), in particular relative to the apoprotein (Supplementary Movie 2).

Non-protein aggregates constitute less than 1% of the total mass and are therefore unlikely to impact the SAXS studies of the *Te*LPOR holoprotein, though the presence of small amounts of protein aggregates among the large particles cannot be ruled out. In the SAXS experiments below, these larger aggregates/ particulates would presumably be removed by the preparative centrifugation (20.000 × *g*, 20 min, 4 °C) immediately prior to the SAXS measurements.

Comparing the MWA-AUC derived holoprotein absorption spectrum to Pchlide solution spectra confirms Pchlide binding to *Te*LPOR (Fig. 1e and Supplementary Fig. 3d, e). In particular, the Pchlide Q$_y$-band of the *Te*LPOR holoprotein (red curve) is blue-shifted compared to Pchlide in TritonX-100 free buffer, likely representing Pchlide aggregates (cyan curve). Differences between *Te*LPOR holoprotein (red curve) and free Pchlide monomers (dark green curve, with Triton X-100) are less obvious due to the noisy MWA-AUC spectrum. Nevertheless, the holoprotein Pchlide Q$_y$ band is slightly red-shifted relative to the Q$_y$ band of free Pchlide monomers (dark green curve) more closely matching the peak position of fully protein bound Pchlide (green curve)[5,25]. Comparing absorption spectra of holoprotein and sedimenting material, we estimated the oligomer distribution for the holoprotein sample (Supplementary Discussion Section 1.2) indicating that 89% of the total protein is monomeric (67% apo; 22% holoprotein), while 11% are constituted by the dimeric holoprotein - in agreement with the SAXS analyses for a similar holoprotein sample (Supplementary Table 2; $c = 0.65$ mg ml$^{-1}$; monomer content from $I(0)$: 85.9%). To test both monomeric and dimeric holoprotein species for activity, we analysed a monomer- and dimer-containing holoprotein sample after illumination by

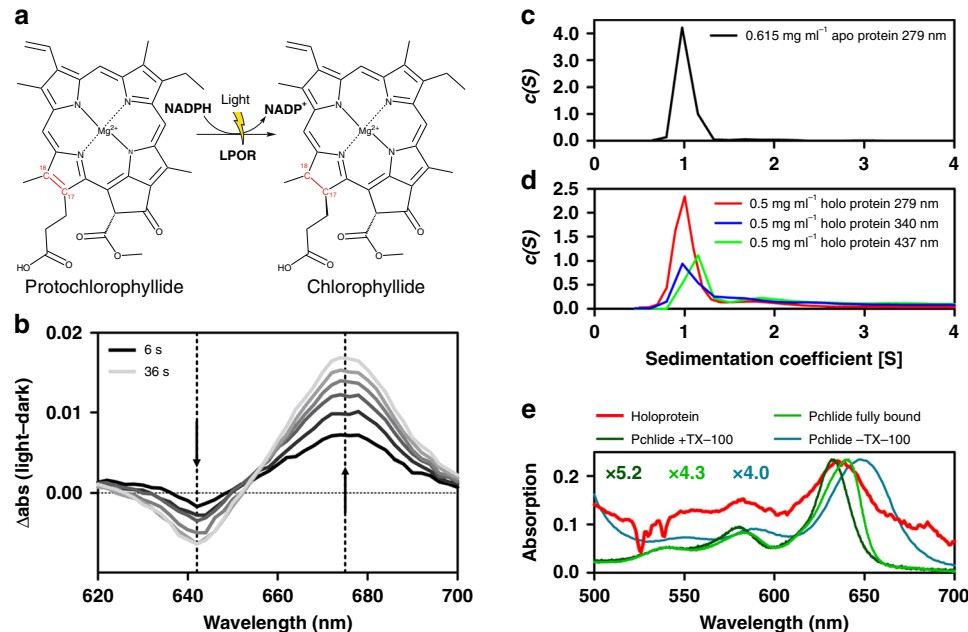

**Fig. 1** Catalyzed reaction, functionality tests and multi-wavelength analytical ultra-centrifugation of TeLPOR. **a** LPORs catalyze the light-dependent reduction of the C17 = C18 double bond (highlighted in red) of protochlorophyllide (Pchlide) to chlorophyllide. **b** Light-dark difference absorption spectra illustrate holoprotein activity. Samples were illuminated with blue light at 6 s intervals (6 to 36 s; shown in shades of grey). The decrease in the Pchlide absorption band (642 nm) and the concomitant increase of the Chlide product band (675 nm) are marked by dashed lines and the direction of the change is indicated by arrows. **c** Sedimentation coefficient distributions c(s) of the predominately monomeric (~1 S) 0.5 mg ml$^{-1}$ apoprotein sample. **d** Sedimentation coefficient distributions c(s) of a 0.5 mg ml$^{-1}$ holoprotein sample at either 279, 340 or 437 nm, showing the relative abundances of monomeric protein (~1 S) and nearby dimer (~2 S) (see also Supplementary Fig. 3c). **e** Absorption spectra of the holoprotein (red line) taken from the first scan at central radial position during analytical ultracentrifugation and absorption spectra of free Pchlide (3.5 µM with (dark green line) and without Triton-X100 (cyan line)), along with the absorption spectrum of fully protein bound Pchlide in reaction buffer (green line). The latter three spectra were recorded on a benchtop spectrophotometer and scaled to yield similar absorption at the Pchlide Q$_y$-band (see scaling factors in the figure). See Supplementary Fig. 3d for full data

MWA-AUC (Supplementary Fig. 3, g, h). Both *c(s)* distributions (Supplementary Fig. 3g) and extracted absorbance spectra (Supplementary Fig. 3h) indicate that the ratio of Pchlide/ NADPH containing holoprotein monomer:dimer increases after illumination from approximately 2:1 to 3-4.5:1, indicative of holoprotein dissociation into monomers (Supplementary Fig. 3g). Unfortunately, we observed only minor Chlide production as discernible from the small Chlide product band at 671 nm (Supplementary Fig. 3h), with the peak position hinting at the presence of Chlide/TeLPOR binary complexes[26]. This assignment is corroborated by the fact that the Chlide product co-sediments with the both the monomer and the dimer. These results suggest that either both holoprotein monomers and dimers are active, or that the Chlide-containing monomers derive from dissociation of the Childe-containing dimers. Given the altered monomer:dimer distribution which we observed for the illuminated holoprotein, we would favor the latter interpretation. For a more detailed discussion see Supplementary Discussion Section 1.2.

**Apo- and holoprotein SAXS studies**. We used small-angle X-ray scattering (SAXS) experiments to obtain structural information of the TeLPOR apoprotein and to get further insights into the structural consequences of ternary complex formation (Supplementary Table 1, Supplementary Figs. 4 and 5). Three independently prepared TeLPOR apoprotein samples as well as one dark-adapted TeLPOR holoprotein sample were analysed. As representative data set for the TeLPOR apoprotein, we selected the data of the apo 2 sample (see Supplementary Discussion Section 1.3 for details). In contrast to the apoprotein, which appears predominantly

monomeric at all tested concentrations (Supplementary Table 2, Supplementary Fig. 6), the holoprotein tends to show increased dimer content at higher concentration (Supplementary Table 2, Supplementary Fig. 6). We therefore used the SAXS data of the holoprotein with highest concentration (5 mg ml$^{-1}$) as representative data set in the following. The direct comparison of representative SAXS data of the TeLPOR apo and holoprotein are shown in Fig. 2. The direct comparison of representative SAXS data of the TeLPOR apo and holoprotein are shown in Fig. 2.

The results of the SAXS data evaluation are compiled in Table 1. Guinier analysis yielded radii of gyration ($R_g$) of 23.0 ± 0.015 Å and 31.6 ± 0.006 Å for the apo and holoprotein, respectively. Corresponding Guinier plots are shown in Fig. 2d. Data points at low *q* values were excluded (highlighted in grey in Fig. 2a, b, d) to alleviate potential small effects due to aggregation. The molecular mass *M* of the apo and holoprotein determined from the Porod volume ($V_p$) was 38,649 Da and 55,895 Da, respectively.

Here, the molecular mass of the apoprotein as calculated from the SAXS data agrees very well with the theoretical molecular mass of a monomer as calculated from the amino acid sequence (*M* = 38,014 Da). The apparent experimentally determined molecular mass *M* for the holoprotein is about 1.5-fold larger than the theoretical molecular mass of a monomer, which indicates a monomer:dimer equilibrium with a dimer content of about 42% (under the assumption that only monomer and dimer contribute to the SAXS data).

**SAXS-guided modelling of the TeLPOR apoprotein monomer**. Prompted by the lack of an LPOR crystal structure, we generated

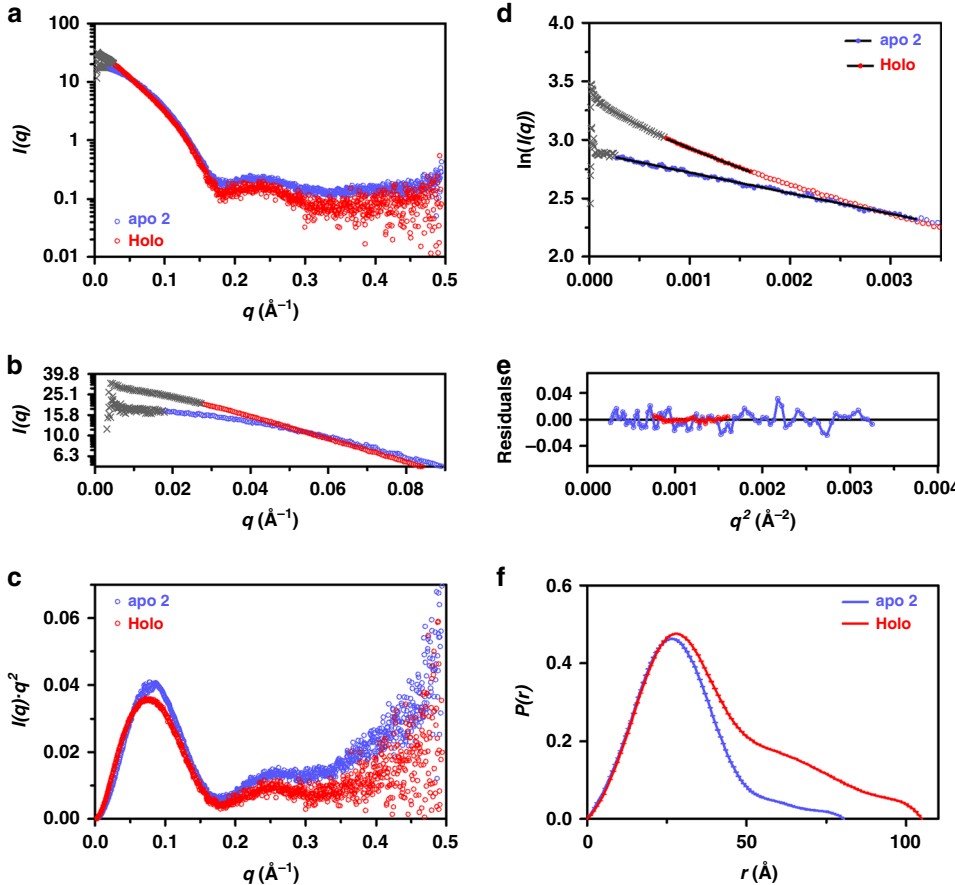

**Fig. 2** Apo and holoprotein *Te*LPOR SAXS data. Representative SAXS data of the *Te*LPOR apo- (apo 2 dataset; light blue, open circles) and holoprotein (5 mg ml$^{-1}$ dataset; red, open circles)(for details see: see Supplementary Discussion Section 1.3). The full datasets of the employed concentration series are shown in Supplementary Figs. 4 and 5. **a** SAXS scattering curve. **b** Scattering curve in the low $q$ range. **c** Kratky plots ($I(q).q^2$ versus $q$). **d** Guinier plots ($\ln(I(q))$ versus $q^2$) and **e** residuals) for $qR_g<1.3$. Open symbols indicate data beyond the Guinier region. **f** Pair distribution function, $P(r)$. Data excluded from Guinier analysis in the low $q$ range is shown as grey crosses in all plots

**Table 1 Evaluation of the SAXS scattering data of the *Te*LPOR apo and holoprotein**

|  | apoprotein | holoprotein |
| --- | --- | --- |
| $I(0)$ (cm$^{-1}$) | 0.013785 ± 0.00005 | 0.109217 ± 0.00037 |
| $M$ from $I(0)^a$ (Da) | 37,096 ± 131 | 52,108 ± 176 |
| [dimer content (%)$^b$] | [0] | [32.4 ± 0.3] |
| $V_p$ (Å$^3$) | 65,730 | 96,680 |
| $M$ from $V_p$ (Da) | 38,649 | 55,895 |
| [dimer content (%)] | [1.7] | [42.0] |
| $R_g$ (Å) from Guinier | 23.0 ± 0.015 | 31.6 ± 0.013 |
| $R_g$ (Å), reciprocal space/ real space$^c$ | 22.9/22.9 ± 0.006 | 32.3/32.5 ± 0.006 |
| $D_{max}$ (Å)$^c$ | 80.5 | 105.4 |
| $M$ from chemical composition | 38,014 | 39,370 |
| $M$ from *ab initio* modelling (Da) DAMMIF/DAMMIN/ GASBORP$^d$ | 35,910/31,110/ 29,498 | 58,153$^e$ |

$^a$determined as described in Supplementary Table 2; $^b$assuming the presence of only monomers and dimers, determined as described in Supplementary Table 2; $^c$derived by using GNOM; $^d$determined by dividing the filtered volume of the DAMAVER generated averaged and filtered model by a factor of two $^e$determined from the GASBORMX model of the dimer as described in $^d$

a homology model for the *Te*LPOR core domain. This model, which is based on a previously published model of the related LPOR of *Synechocystis* sp.[14], lacks 31 C-terminal residues and does not cover the 20 residue N-terminal His$_6$-tag. Circular dichroism spectroscopic data[27] and secondary structure predictions (Supplementary Fig. 7) indicate the presence of an α-helix within the missing C-terminal segment. Based on this assumption, we generated different C-terminally extended *Te*LPOR homology models (Fig. 3a). All models were cross validated against experimental SAXS data of the apoprotein (Fig. 3b, c) and Molecular Dynamics (MD) simulations were performed to evaluate the structural stability of the models (Fig. 3d–f). For a more detailed discussion see Supplementary Discussion Section 1.4.1. Additionally, we considered different core domain structure variants (Supplementary Discussion Section 1.4.2; Supplementary Fig. 8) and used ensemble optimization modelling (EOM) of SAXS data to assess, whether the missing N- and C-terminal residues adopt a flexible, unordered conformation (Supplementary Discussion Section 1.4.3; Supplementary Fig. 9). EOM was used as well to address the possibility that certain internal segments adopt a flexible/disordered conformation in *Te*LPOR. Here, in particular, a 25–30 residue long sequence stretch (here called LPOR insertion loop), which is not present in related short-chain dehydrogenases[14], was modelled as either fully or partially flexible (see Supplementary Discussion Section 1.4.4; Supplementary Fig. 9c–h). Details about EOM modelling, simulations

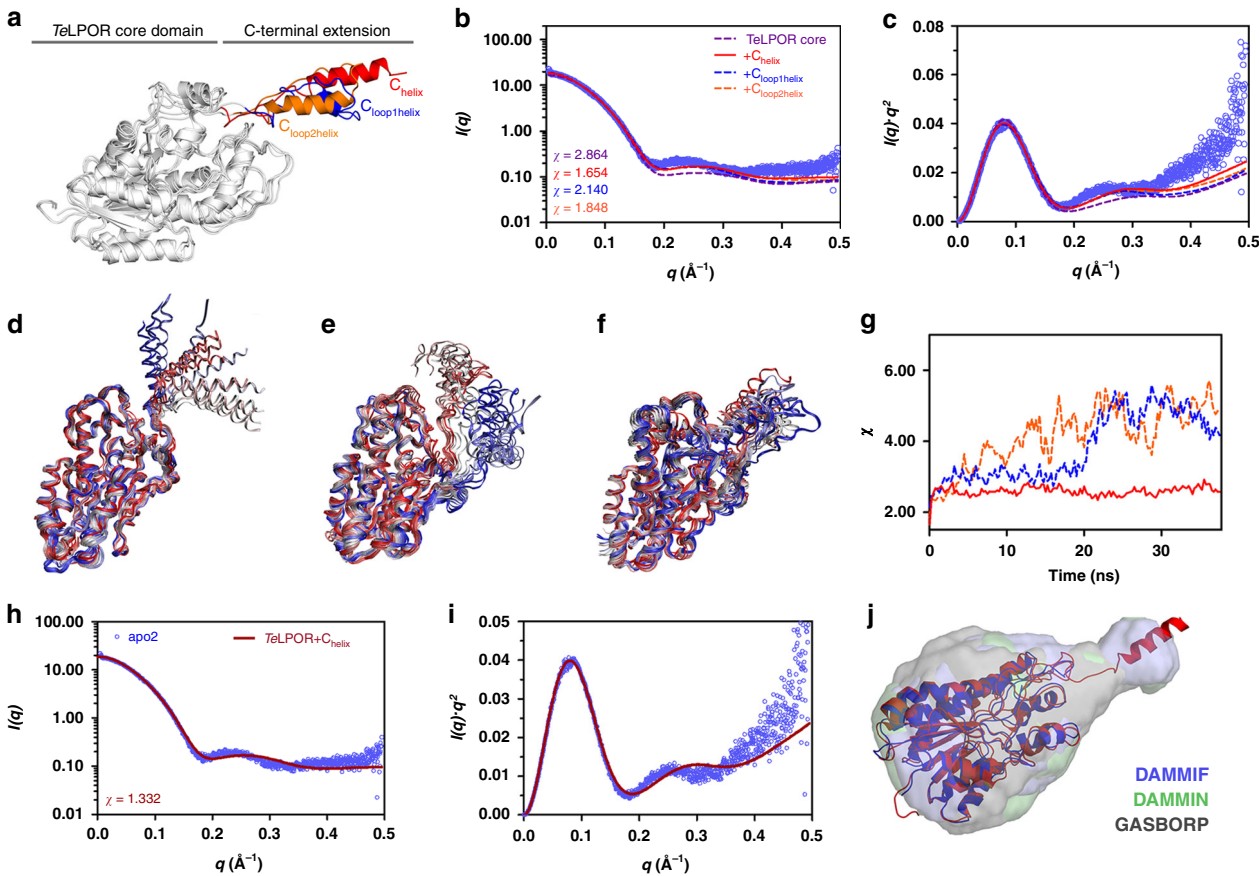

**Fig. 3** SAXS-guided modelling of the *Te*LPOR apoprotein monomer. **a** Three different, C-terminally extended *Te*LPOR models (*Te*LPOR-C$_{helix}$ (red), *Te*LPOR-C$_{loop1helix}$ (blue) and *Te*LPOR-C$_{loop2helix}$ (orange) and CRYSOL-based model evaluation, showing **b** the SAXS scattering curve and **c** the Kratky plot ($I(q).q^2$ versus $q$), with the respective theoretical scattering curve superimposed (solid and dashed lines; color coded as in **b**) on the experimental scattering data of the apoprotein (light blue, open circles). To enable the direct comparison between the models, no constant was subtracted during the CRYSOL fit. Superimposition of selected snapshots of the simulation trajectories of **d** *Te*LPOR-C$_{helix}$, **e** *Te*LPOR-C$_{loop1helix}$ and **f** *Te*LPOR-C$_{loop2helix}$. **g** CRYSOL-derived $\chi$ value for the fit of the theoretical scattering curves of 150 molecular dynamics snapshots against of the experimental scattering data of the *Te*LPOR apoprotein (color coded as in **b**). **h** CRYSOL-based model evaluation for the best C-terminally extended *Te*LPOR model (*Te*LPOR-C$_{helix}$), showing **h**, the SAXS scattering curve and **i** the corresponding Kratky plot, with the theoretical scattering curve of the *Te*LPOR-C$_{helix}$ model (solid dark red line) superimposed on the experimental scattering data of the apoprotein (light blue, open circles). Compared to **b**, the fit of the theoretical scattering curve was improved by constant subtraction, which accounts for systematic errors due to mismatched buffers. **j** SITUS-derived envelope function, obtained from averaged and filtered DAMMIF- (blue), DAMMIN- (green) and GASBORP (grey)-derived *ab initio* models (transparent surface) superimposed on the best C-terminally extended *Te*LPOR model (*Te*LPOR-C$_{helix}$, red cartoon). For comparison also the *Te*LPOR core domain model (blue cartoon) is shown

and model validation can be found in the Supplementary Information (see Supplementary Discussion 1.4). Overall, our analyses indicate, that the previously presented *Te*LPOR core domain models (residues 1–285)[1,10,13–17] are insufficient to explain our apoprotein SAXS data. Modelling the missing 31 C-terminal amino acids as protruding α-helix (*Te*LPOR-C$_{helix}$ model, Fig. 3a) markedly improves the fit (Fig. 3b, c, h, i; compare $\chi$ value for *Te*LPOR core model ($\chi = 2.864$) and *Te*LPOR-C$_{helix}$ ($\chi = 1.654$)). All of the tested models possessing alternative core-domain structures or flexible/disordered terminal and/or internal segments yielded worse $\chi$ values. MD simulations show that the structure of the core domain of the *Te*LPOR-C$_{helix}$ model is very stable (Fig. 3d, Supplementary Fig. 10), whereas the C-terminal protruding helix appears mobile (Fig. 3d). This movement does, however, not translate to markedly worse $\chi$ values for the fit against the experimental apoprotein SAXS data (Fig. 3g; red line). To further validate our apoprotein models, we performed *ab initio* modelling[28] (see Methods, Small-angle X-ray scattering (SAXS) for details; Supplementary Fig. 11). The overall shape of the obtained molecular envelopes resembles a bowling-pin

(Fig. 3j) appearing to consist of a larger and smaller sub-domain, which is corroborated by the corresponding $P(r)$ plot (Fig. 2f, blue line).

Superimposition of the best C-terminally extended *Te*LPOR homology model (*Te*LPOR-C$_{helix}$) to the corresponding low-resolution envelopes allows for a placement of the conserved LPOR core Rossmann-fold within the larger subdomain, with the C-terminal helix covering the small subdomain (Fig. 3j, red cartoon). In contrast, the corresponding *Te*LPOR core domain model (Fig. 3j, blue cartoon) is insufficient to fill-out the envelope shape completely.

**SAXS-guided modelling of the *Te*LPOR holoprotein dimer.** To obtain an atomic model of the dimeric *Te*LPOR holoprotein, we used restraint-free homo-multimer docking by employing the ClusPro webserver (https://cluspro.bu.edu)[29]. As input monomer models, we used the model of the *Te*LPOR core domain (Fig. 3j, blue, cartoon) or the C-terminally extended *Te*LPOR-C$_{helix}$ model (Fig. 3j, red cartoon). The resulting dimer

models were subsequently scored by comparison against the experimental SAXS data. The best model was selected based on the lowest $\chi$ value of the fit of the theoretical scattering curve of a monomer/dimer mixture by employing the OLIGOMER algorithm (Fig. 4a) (see also Supplementary Discussion Section 1.5, Supplementary Table 3, Supplementary Figs. 12 and 13 for details). For model selection, we also considered distance constraints provided by a recent cross-linking study of the LPOR A enzyme of *Arabidopsis thaliana* (*At*LPOR A)(Supplementary Table 3)[20]. Consistently, all dimer models containing the C-terminal α-helical extension provided a better fit to the experimental SAXS data (compare Supplementary Figs. 12 and 13). For the best dimer model (dimer 4b; see Supplementary Discussion Section 1.5.1, Supplementary Table 3) we performed an MD simulation to validate its stability (Supplementary Fig. 14). For a detailed discussion of the corresponding results refer to the Supplementary Information (Supplementary Discussion Section 1.5.2, Supplementary Figs. 14–18, Supplementary Table 4). While during the simulation, some minor rearrangements take place, i.e. with regard to the placement of the C-terminal helix (Supplementary Fig. 14d), the fit of the theoretical scattering curves of MD trajectory snapshots against the experimental SAXS data did not improve markedly over the simulation time (Supplementary Fig. 14b, Supplementary Fig. 15). Since the mode of interaction, as well as the overall dimer structure did not change dramatically during the simulation, we present in the following only the data for the initial ClusPro-derived dimer 4b.

The overall mode of dimerization, driven by both hydrophobic (Supplementary Figs. 16 and 17) and electrostatic interactions (Supplementary Fig. 18) remains stable in MD simulations. The interface between the subunits is hereby largely constituted by highly conserved residues on both the Rossmann-fold core domain and the C-terminal α-helix (Fig. 4b). Compared to the MD simulation performed for the *Te*LPOR-C$_{helix}$ apoprotein model (Fig. 3d), the observed mode of dimerization seems to stabilize the movement of the C-terminal helix, as evidenced by a reduced root mean square fluctuation (RMSF) of the corresponding segment (Fig. 5a; region highlighted in green). Importantly, the OLIGOMER-derived fit of the theoretical scattering curve of

the monomer/dimer mixture, calculated from the dimer 4b model, and the experimental SAXS data yields a moderately good $\chi$ value of $\chi = 1.637$ (Fig. 5b, c).

Finally, we reconstructed a molecular envelope of the holoprotein dimer (see Methods, Small-angle X-ray scattering (SAXS) for details; Fig. 5d). The best ClusPro-derived dimer model (dimer 4b; see Supplementary Discussion Section 1.5.1, Supplementary Table 3) accounts well for the SAXS-derived *ab initio* holoprotein dimer envelope. In this dimer model the protruding C-terminal helix directly interacts with the opposite subunit thereby stabilizing a domain-swapped LPOR dimer (Fig. 5d; highlighted in green). Moreover, two of the monomer apoprotein envelopes (Fig. 3j) fit well into the holoprotein dimer envelope (Supplementary Fig. 19).

**C-terminal truncation of *Te*LPOR**. To probe the role of the C-terminal extension for the structure and activity of *Te*LPOR, we constructed truncated versions of *Te*LPOR by introducing a stop codon at three different sites within the C-terminal segment not covered by previous homology modelling studies[14,17] (P272, V279 and A302)(Fig. 5e). This resulted in the variants *Te*LPOR-Δ51, *Te*LPOR-Δ44 and *Te*LPOR-Δ21 shortened by 51, 44, and 21 amino acids compared to *Te*LPOR wild type. We produced all variants heterologously in *E. coli* (Supplementary Fig. 20a), and attempted to purify them by IMAC. This, however, yielded low amounts of the target protein, with the samples containing larger amounts of unknown co-purified protein (Supplementary Fig. 20b–d). At best, *Te*LPOR-Δ21 was obtained at about 50% purity (Supplementary Fig. 20b), hampering further biophysical characterization. Nevertheless, we tested light-dependent Pchlide turnover using both cell-free cell extracts (Fig. 5f) and purified protein (Supplementary Fig. 21). Interestingly, neither of the variants showed any detectable light-dependent Pchlide turnover. This might indicate that the C-terminal protein segment is important for the proper folding of *Te*LPOR. However, further variants, i.e. with more conservative, shorter, truncations, would have to be studied to ascertain the role of the C-terminal extension for the function and dimerization of *Te*LPOR.

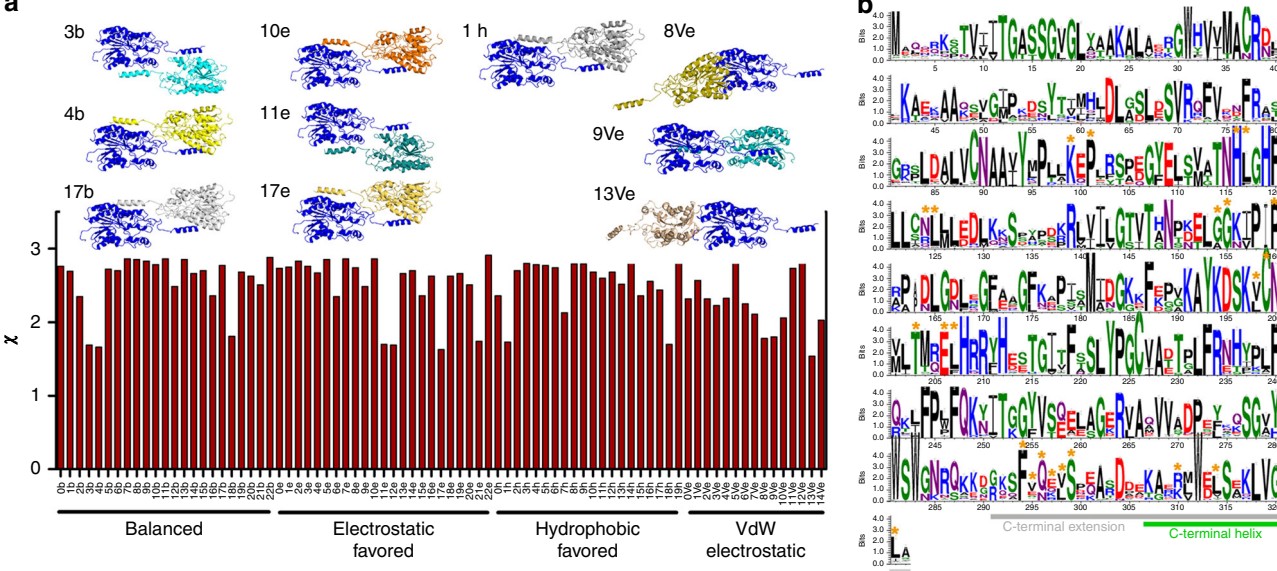

**Fig. 4** SAXS-guided modelling of the *Te*LPOR holoprotein dimer and LPOR residue conservation. **a** Evaluation of the ClusPro-derived *Te*LPOR-C$_{helix}$-based dimer models with regard to $\chi$ for the fit against the experimental scattering data of the holoprotein using the program OLIGOMER[55]. The ten best ClusPro-derived dimer models are shown above the plot (Supplementary Table 3). **b** Weblogo 3[74] generated sequence logo, illustrating the conservation of residues within the LPOR family. Residues, which contribute to dimerization are marked by an orange asterisk

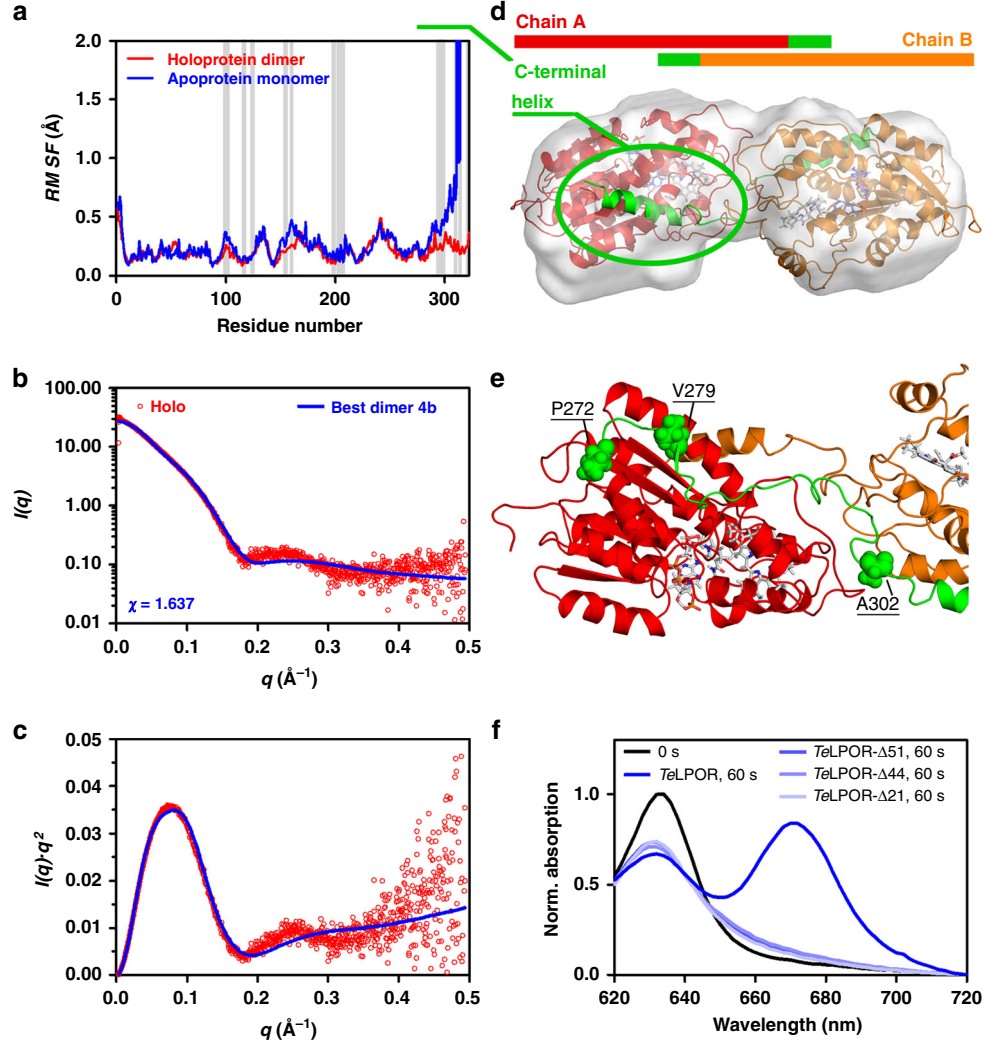

**Fig. 5** SAXS-guided modelling of the *Te*LPOR holoprotein dimer and *Te*LPOR truncation. **a** Root mean square fluctuation (RMSF) per residue obtained from the MD simulations of the *Te*LPOR apoprotein monomer (blue) and the *Te*LPOR holoprotein dimer (red). Grey vertical bars mark dimer interface residues. **b**, **c** SAXS scattering curve and Kratky plot ($I(q).q^2$ versus $q$) showing the experimental scattering data of the holoprotein (red, open circles) and the fit of the OLIGOMER-derived theoretical scattering curve of a monomer/dimer mixture of dimer 4b (blue, solid line). **d** Averaged and filtered GASBORMX-derived *ab initio* bead model of the dimeric holoprotein complex as SITUS-derived envelope (transparent, grey surface), superimposed with the best *Te*LPOR holoprotein dimer model (dimer model 4b; Supplementary Table 3) Subunits colored as indicated above the figure, with the protruding C-terminal α-helix in green). **e** One subunit of *Te*LPOR (red cartoon), with 51 C-terminal amino acids shown as green cartoon, illustrating the truncation positions (P272, V279 and A302; as green spheres) to produce the variants *Te*LPOR-Δ51, *Te*LPOR-Δ44 and *Te*LPOR-Δ21. **f** Light-dependent Pchlide turnover, analysed using cell-free cell extracts of *E.coli* BL21(DE3) cells producing *Te*LPOR-Δ51, *Te*LPOR-Δ44 and *Te*LPOR-Δ21 and wild type *Te*LPOR. Sample identity as indicated in the Figure

## Discussion

Due to the lack of an LPOR X-ray structure little is known about the tertiary structure of the enzyme, except that it likely contains a Rossmann fold[1], as inferred from sequence similarity to short-chain dehydrogenase/reductase (SDR) enzymes. Likewise, all presented homology models[1,10,13–17], built based on SDRs, remain experimentally unvalidated, and do not cover the full-length LPOR sequence, i.e. missing residues at the C-terminus that are not covered by the template used for model building. Our SAXS data for the *Te*LPOR apoprotein provides strong evidence that the current *Te*LPOR core (Rossmann-fold) homology model, although partly accounting for our SAXS data, has to be extended (Fig. 3). Here, modelling the missing 31 C-terminal amino acids as protruding α-helix, which is in line with previous CD spectroscopic studies[27] and secondary structure predictions[15], yields the best fit to our apoprotein SAXS data. This structural

architecture, which is stably maintained during MD simulations (Fig. 3d, g and Supplementary Fig. 10b), is corroborated by the bowling-pin like shape of the corresponding SAXS-derived *ab initio* models (Fig. 3j). Interestingly, while the C-terminal helix appears to move freely during MD simulations (Fig. 3d), this movement does not to translate to markedly worse χ values for the fit against the experimental SAXS data (Fig. 3g). This indicates that a freely moving, yet folded, C-terminally protruding helix is accommodated by our apoprotein SAXS data. Apart from providing a glimpse into the full-length architecture of LPORs, our SAXS and MWA-AUC data unequivocally shows that the *Te*LPOR apoprotein is monomeric in solution. This is in contrast to most other SDR enzymes, which either form homotetramers or homodimers[30]. The largest oligomerization interface, conserved among SDRs, is the so-called Q-axis interface constituted by two long α-helices (commonly called αE and αF), which form an

antiparallel four-helix bundle in the dimer[31]. Dimerization predominately occurs via a hydrophobic patch on the surface of the two helices[32]. Compared to oligomeric SDRs of the hydroxysteroid-dehydrogenases family (HSDs), like 7α-HSDs[33], 3α,20β-HSDs[32] and 17β-HSDs[34], TeLPOR possesses a much less well pronounced hydrophobic surface in the corresponding region (Supplementary Fig. 22), which is moreover interspersed with charged residues (Supplementary Fig. 18). This in turn might account for the weak dimerization tendency of the protein in its apo form.

SAXS and MWA-AUC data of the TeLPOR holoprotein independently verify that Pchlide and NADPH binding result in dimerization of TeLPOR. Protein-Protein docking of two full-length TeLPOR monomers produced a set of very similar dimer models (Fig. 4a, Supplementary Figs. 12 and 13), which account well for our SAXS data (Fig. 5b–d). Interestingly, in our best-fitting TeLPOR dimer model, dimerization occurs via the aforementioned Q-axis interface of one subunit and the C-terminally protruding α-helix of the opposite monomer (Fig. 5d) in a domain swapped manner, with the C-terminal helix concealing most of the hydrophobic patch on the surface of the protein (Supplementary Fig. 17). This interaction finds support in an independent cross-linking study of the LPOR A of A. thaliana[20]. Here, the authors observed crosslinks between K129 localized on the surface of the conserved Rossmann-fold) and K390 (within the C-terminal extension) for the AtLPOR A apoprotein, which were absent in the corresponding holoprotein sample. In our TeLPOR holoprotein dimer model, the residue equivalent to K390 in AtLPOR A (K307 in TeLPOR) is found buried within the holoprotein dimer interface, and would thus only be accessible for cross-linking in the apoprotein. Mechanistically, the here suggested dimerization mode via the C-terminally protruding helix is reminiscent of the structure of the monomeric Porcine testicular carbonyl reductase, where a 41-residue insertion, not found in oligomeric SDRs, forms an all-helix subdomain that packs against the Q-axis interface hindering dimerization[31]. It is thus tempting to speculate that in TeLPOR and other oligomeric SDRs, a similar interface on the Rossmann-fold core domain is involved in dimerization, and that in monomeric SDRs the same interface is shielded by additional structural elements. Moreover, the here proposed TeLPOR dimer interaction, i.e. with the protruding C-terminal helix stabilizing the domain-swapped LPOR dimer, is reminiscent of the subunit interactions seen for structurally dissimilar DPOR enzyme, in which the Pchlide molecule held by the N- and B-subunits interacts with the C-terminal α-helix of B'-subunit (in the other symmetric unit) forming the catalytic (N-B)₂ hetero-tetramer[35].

Irrespective of the mode of dimerization, the question immediately arises how substrate/cofactor binding within the TeLPOR active site structurally mediates dimerization. First, NADPH and Pchlide binding might result in a reorganization of the potential dimerization interface (i.e. the weakly hydrophobic Q-axis interface of TeLPOR), resulting in increased surface hydrophobicity, which would then drive dimerization. In this case, one would expect that dimerization, as in other SDRs, would occur via the Q-axis interface resulting in an antiparallel Rossmann-fold dimer with protruding C-terminal α-helices. To test this possibility, we generated a full-length TeLPOR dimer model dimerized via the Q-axis interface as found in dimeric 7α-HSDs (Supplementary Fig. 23a)[33]. The theoretical scattering curve of this dimer model does not fit well to the experimental SAXS data of the holoprotein (Supplementary Fig. 23b, c; $\chi = 2.917$), i.e. compared to our above presented best ClusPro-derived dimer model (dimer 4b; Fig. 5b, c, Supplementary Fig. 23b, c; $\chi = 1.637$), rendering this possibility unlikely. Secondly, flexible regions of the protein might become ordered upon ternary complex formation, which is a known feature of other SDR enzymes[33,36]. This in turn might present a novel interface for dimerization, which is absent in the apoprotein, thereby driving dimerization. This model could not be evaluated as it assumes substantial conformational changes in the monomer due to Pchlide/NADPH binding, which we cannot assess as our starting TeLPOR monomer model already contained bound Pchlide and NADPH (see Supplementary Discussion Section 1.4.1). Please note, however, that our disorder prediction for TeLPOR did not detect any potential disordered regions other than the LPOR insertion loop and the C-terminal extension. Likewise, dimerization might induce further larger scale conformational changes within the monomer, which are not accommodated by the ClusPro rigid-body docking methodology[29]. Last but not least, the conformation or flexibility of the C-terminally protruding α-helix of TeLPOR could present a switch that drives dimerization. This notion is supported by our docking studies and the observation that removal of the C-terminal extension resulted in poorly expressed or inactive protein (Fig. 5f, Supplementary Figs. 20b–d and 21). However, to ascertain the role of the C-terminal extension for TeLPOR function and dimerization further studies would be needed. For the LPOR of Pisum sativum alanine substitions within the C-terminal extension resulted in loss of LPOR activity[15], which corroborates the importance of this structural region for LPOR function.

With regard to those possibilities, we propose the following model (Fig. 6) that accommodates all of our SAXS and modelling data as well as previous observations regarding the structure and function of the LPORs: In absence of NADPH and Pchlide, TeLPOR is monomeric, with the C-terminal helical extension

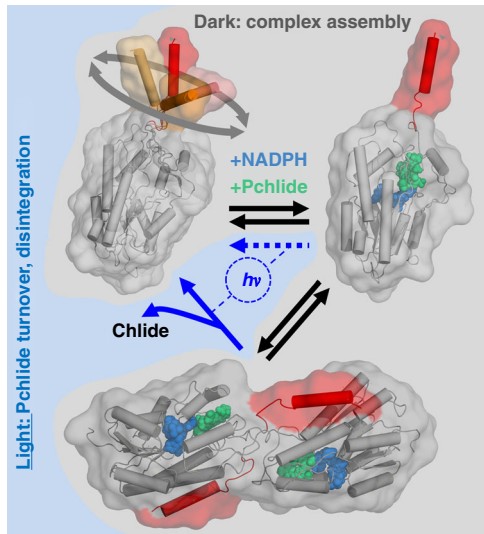

**Fig. 6** Model illustrating the proposed structural changes occurring in TeLPOR due to holoprotein formation and light-dependent Pchlide turnover. For a detailed description of the model refer to the main text. Without Pchlide and NADPH, TeLPOR is monomeric, with the C-terminal α-helical extension moving freely (illustrated by four different helical conformations extracted from the corresponding MD trajectory; orange to red cartoon). Assembly of the holoprotein complex in the dark might result in rigidification of the protein, which in turn allows for dimerization, mediated by the C-terminal extension, the Q-axis interface and the active site surface patch. Light-dependent conversion of Pchlide to Chlide, followed by product release would then trigger the dissociation of the dimer. The dashed blue arrow indicates the possibility that also the monomeric holoprotein shows light-dependent activity, as inferred by MWA-AUC. Pchlide and NADPH are shown as green and blue spheres, respectively. The molecular surface of the models are shown as transparent grey surface

likely being flexible (as indicated by MD simulations; see above). This would hinder dimerization via this structural element in the apo-form. Binding of NADPH and Pchlide to the protein, induces a conformational change in the protein monomer[25], which might involve rigidification of the C-terminal extension. Such a conformational change, which is not directly assessed by our modelling strategy, could for example be triggered by an interaction between Pchlide or the substrate-binding site and the loop region that connects the Rossmann-fold core domain and the C-terminal helix. Interestingly, in *E. coli* 7α-HSD the C-terminus, which, however, is much shorter as in *Te*LPOR, undergoes such a conformational change upon ternary complex formation[33]. This change in flexibility in turn might allow for formation of the dimeric photoactive holoprotein. This model would also account for the light-induced disintegration of prolamellar bodies[37,38], i.e. with light-dependent Pchlide turnover and subsequent Chlide release resulting in the reversal of the proposed processes (Fig. 6) which drives the disintegration of prolamellar bodies. The latter step is also supported by the above presented MWA-AUC studies performed for the illuminated *Te*LPOR holoprotein that showed increased monomer content after illumination (Supplementary Fig. 3g).

Please note that this mechanistic proposal is solely based on the here presented consensus *Te*LPOR apoprotein monomer and holoprotein dimer models as well as simulation data. Those models, selected from several likely alternatives, are best-fitting to our experimental SAXS data, they are physically feasible and remain stable in MD simulations. Given the low resolution nature of SAXS data, we, however, cannot not rule out that other structural models, not covered by our alternative modelling approaches, exist, that would equally well account for the presented SAXS data. High-resolution X-ray or NMR- structures will be needed to address this issue unequivocally.

In summary, our study provides to the best of our knowledge the first glimpse into the structural architecture of the yet structurally uncharacterized LPOR enzyme family, paves the way for further biophysical characterization, mutational and computational studies, and advances our structural understanding of this important class of enzymes.

## Methods

**Strains and plasmids**. The gene coding for the LPOR of *Thermosynechococcus elongatus* BP-1 (UniProt ID: Q8DLC1) (*Te*LPOR) was custom synthesized (ThermoFisher Scientific/Life technologies, Waltham, MA, USA) without codon optimization. The synthetic gene was appended with a 5′- *Nde*I and a 3′- *Sal*I restriction endonuclease recognition site, and sub-cloned from the synthesis vector into pET28a (Merck/Novagen, Darmstadt, Germany) as expression vector. This results in an in frame fusion of a 20 amino acid N-terminal hexa-histidine (His₆) tag (tag sequence: MGSSHHHHHHSSGLVPRGSH). C-terminally truncated variants of *Te*LPOR were generated by QuikChange-PCR by introducing a stop codon at the codon positions 272, 279 and 302 using the oligonucleotides listed in Supplementary Table 5. QuikChange-PCR was performed by employing Phusion DNA Polymerase (Thermo fisher Scientific, Ulm, Germany) according to the instructions provided by the Supplier of the QuikChange Site-Directed Mutagenesis kit (Agilent, Santa Clara, CA, USA). All final constructs were verified by sequencing (SeqLab, Göttingen, Germany).

**Heterologous gene expression and protein purification**. The *Te*LPOR encoding gene was expressed as His₆-tagged fusion protein in *E. coli* BL21(DE3). Expression cultures were grown in 5 l non-baffled Erlenmeyer flasks (500 ml modified auto-induction media[39] supplemented with kanamycin (50 μg ml⁻¹) for plasmid maintenance). All cultures were grown initially for 2 h at 37 °C under constant agitation (250 rpm). Subsequently the temperature was decreased to 15 °C and cells were grown for additional 48 h. Expression cultures were harvested by centrifugation (30 min, 6.750 x*g*, 4 °C) and resuspended in 20 mM Tris/HCl, 500 mM NaCl, 20% (w/v) glycerol, pH 7.5. Cell disruption was achieved by passing the cell suspensions (10% (w/v) wet cells) five times through an EmulsiFlex-C5 high-pressure homogenizer (AVESTIN, Ottawa, Canada) at a pressure of 1000 bar. Cell debris and unbroken cells were removed by centrifugation (30 min, 30.310 x*g*, 4 °C). The recombinant *Te*LPOR protein was purified by immobilized metal ion affinity chromatography (IMAC) as described previously[40]. Immediately after IMAC, the protein sample was desalted using a Sephadex™ G25 column (560 ml column volume, XK50/30, GE Healthcare Life science, VWR International GmbH, Langenfeld Germany). Protein samples were concentrated to a concentration of at least 1 mg ml⁻¹ using a Nano-sep® centrifugal concentrator unit (molecular weight cutoff 10.000 Da)(Pall, VWR International GmBH, Darmstadt, Germany) and subsequently further purified by preparative size-exclusion chromatography (Superdex™ 200, XK16/60, GE Healthcare Life science, VWR International GmbH, Langenfeld, Germany) employing 20 mM Tris/HCl pH 7.5 buffer supplemented with 500 mM NaCl and 20% (v/v) glycerol as eluent. All samples were flash frozen in liquid nitrogen and stored at −20 °C until further use.

**Protochlorophyllide (Pchlide) production and purification**. Pchlide was produced using *Rhodobacter capsulatus* ZY5[41]. The strain was cultivated in VN-Medium (10 g/l yeast extract, 5.7 μM K₂HPO₄, 2 μM MgSO₄, pH 7.0) in the dark at 30 °C under constant agitation (130 rpm). Production cultures were inoculated to an OD₆₆₀ₙₘ of 0.01 and grown under micro-aerobic conditions (culture volume 50% of the flask volume; non-baffled Erlenmeyer flasks). The secreted Pchlide was adsorbed to hydrophobic polyurethane (PU) cubes (edge length: 1 cm) which were added to the cultures during cultivation. After 24 h the cubes were withdrawn. Cells were removed by washing with tricine buffer (10 mM tricine pH 7.5). Subsequently, Pchlide was extracted from the cubes with 100% methanol and the extract was filtered (cellulose acetate filter, pore size 0.45 μm). Pchlide was purified by column chromatography using an ÄKTAbasic™ FPLC system (GE Healthcare, Solingen, Germany) using C-18 solid-phase extraction (SPE) material (Sep-Pak®, Waters, Milford, MA, USA) filled into an ECOPLUSSR TAC15/500LGO-SR-2 column (75 ml CV)(YMC Europe GmbH, Dinslaken, Germany). To facilitate binding, the filtered Pchlide methanol extract was diluted to a final concentration of 40% (v/v) methanol with tricine buffer. The SPE column was equilibrated with methanol: tricine buffer (40% (v/v) methanol), the Pchlide extract was loaded, and the column was washed using the same methanol:tricine buffer mixture. To separate carotenoids and other unwanted pigments from Pchlide, the methanol concentration was increased stepwise to 50% (after two CV) and 60% (after 25 CV). Finally, Pchlide was eluted using a methanol:tricine buffer ratio of 75:25. The obtained purified Pchlide eluate was diluted with tricine buffer to a final methanol concentration of ~25%. Subsequently, Pchlide was extracted with fresh diethyl ether by liquid-liquid extraction. The resulting diethyl ether Pchlide extract was dried with MgSO₄, the ether was evaporated using a rotary evaporator (Rotavapor® R-100, Büchi, Flawil, Switzerland), and the dried sample was stored under argon atmosphere at −20 °C in the dark.

**Sample preparation for SAXS and MWA-AUC**. *Te*LPOR apoprotein samples were prepared by concentrating a freshly prepared IMAC and SEC purified protein sample starting from a sample of about 0.5 mg ml⁻¹ using a Nanosep® centrifugal concentrator unit (molecular weight cutoff 10,000 Da)(Pall, VWR International GmBH, Darmstadt,, Germany). Samples of defined concentration were removed during the concentration process and the corresponding flow-through was collected as buffer reference for SAXS measurements. The *Te*LPOR holoprotein was prepared by incubating the apoprotein (0.44 mg ml⁻¹; 11 μM) with 11 μM Pchlide and 11 μM NADPH at 25 °C in the dark. For Pchlide, a molar extinction coefficient of 23.95 mmol⁻¹ cm⁻¹[42,43] was used. All subsequent steps were carried out in the dark or under dim green safety light to avoid Pchlide turnover. The reconstituted holoprotein sample was concentrated as described above for the apoprotein samples. Samples of defined concentration were used for SAXS analyses employing the corresponding buffer flow-through as reference. All SAXS samples were centrifuged at for 20 min, 21.000 x *g* at 4 °C to remove larger aggregates and particulate material. Identically prepared samples were used for AUC experiments.

**MWA-AUC experiments and data analysis**. *Te*LPOR apoprotein and holoprotein samples at 1.23 mg ml⁻¹ and 1 mg ml⁻¹, respectively, were freshly diluted with reaction buffer to half the concentration and filled into custom-produced titanium centerpieces with sapphire windows and optical path lengths of 20 mm. Additionally, measurements of undiluted samples were performed in corresponding measurement cells of 12 mm. Upon inserting the cells into the rotor, optical alignment along the centrifugal field is ensured by the application of a custom-made cell alignment tool (Nanolytics Instruments, Potsdam, Germany). Sedimentation velocity experiments were carried out on a Nanolytics Instruments MWA Analytical Ultracentrifuge using multiwavelength absorbance (MWA) optics at 20 °C and an angular velocity of 50 krpm. The MWA detector records full spectra by simultaneously acquiring the absorption at 2048 wavelengths between 185-894 nm for each radial scan[44].

The data for single wavelengths selected from the MWA data sets were analysed with the standard *c(s)* model in SEDFIT version 15.01b to generate a diffusion-corrected sedimentation coefficient distribution with a weight-averaged frictional ratio. For parametrization refer to Supplementary Discussion Section 1.2. The *c(s)* distributions show the signal amplitudes of all species with a given sedimentation coefficient *s* absorbing at the respective wavelength and enable the identification of the oligomeric states of proteins from their apparent molar masses. Additionally, complete MWA data sets were analysed using SEDANAL version 6.82 with the wide distribution analysis (WDA) method providing a non-diffusion corrected

sedimentation coefficient distribution, free of assumptions, for extracting full absorption spectra of specific populations within a given $s$ value range[45]. Deconvolution of component concentration profiles from MWL data sets were performed using SEDANAL[45] and Ultrascan 3[46]. All plots of AUC raw data, best fits and residuals were created with the software GUSSI, which can be downloaded from the MBR Software Page (http://biophysics.swmed.edu/MBR/software.html). Data plots of $c(s)$ and distributions were created by in-house developed software. Movies of sedimentation velocity experiments were generated by LabView-based MWL data viewer written by Dirk Haffke (Universität Konstanz) (http://wiki.bcf2.uthscsa.edu/openAUC/wiki/WikiStart).

**SAXS experiments and data analysis**. SAXS experiments were performed at beamline BM29[47] at the European Synchrotron Radiation Facility (ESRF, Grenoble, France) using 12.5 keV X-ray radiation with a wavelength of 0.992 Å, and a PILATUS 1 M 2D detector. All measurements were carried out at 10 °C. For each *Te*LPOR protein sample (apo and holoprotein), 3-4 samples with concentrations between approx. 0.5 mg ml$^{-1}$ and 7 mg ml$^{-1}$ were measured in 20 mM Tris/HCl buffer pH 7.5 supplemented with 500 mM NaCl and 20% (v/v) glycerol. The exact protein concentrations of the measured LPOR samples are listed in Supplementary Tables 1 and 2. All holoprotein samples were prepared in the dark. The samples were continuously purged through a 1 mm quartz capillary at a flow rate of 2.3 µl/s. During the SAXS measurement a camera monitors sample flow through the quartz capillary employing a cold-light source for illumination. To avoid excitation of the *Te*LPOR holoprotein samples by the cold-light source, a filter was employed that blocks incident light outside the 275–375 nm spectral window. The buffer reference was measured before and after each protein sample. For each sample/reference ten frames with an exposure time of 3 seconds each were recorded and merged. No radiation damage was observed. The data was scaled by the protein concentration and extrapolated to infinite dilution. Scattering data was analysed employing the ATSAS software package[28]. SAXS data was inspected visually for the presence of aggregation based on the Guinier-Plot. $I(0)$ (in cm$^{-1}$) and the Porod volume ($V_p$), were used to estimate the molecular mass $M$ of the scattering particle. The latter was calculated with DATPOROD. $M$ from $V_p$ was obtained by multiplication of $V_p$ with the reported protein density of 0.588 g ml$^{-1}$[28]. For the final apoprotein dataset (apo 2), data obtained for low and high concentration samples was merged (see Supplementary Tables 1 and 2). Lower concentration data was used for the smaller $q$-range, while the data at higher concentration was used for the high $q$-range. As final dataset for the *Te*LPOR holoprotein, the SAXS data of a high concentration sample (5.0 mg ml$^{-1}$) was used directly. The distance distribution function $P(r)$ was determined using the program DATGNOM[48]. *Ab initio* models of the monomeric apoprotein were built employing the programs GASBORP (fitting the real-space $P(r)$ function)[49], DAMMIF[50] and DAMMIN[51]. For each LPOR dataset 20 *ab initio* bead models were generated. The resulting *ab initio* models were averaged using DAMAVER. Overall, DAMMIN, DAMMIF and GASBORP yielded very similar models (Fig. 3j) with SUPCOMB derived normalized spatial discrepancies (NSD) between 0.482 (DAMMIN models vs. DAMMIF model) and 0.531 (GASBORP vs. DAMMIF model). The corresponding averaged and filtered models generated by the individual programs were very similar, i.e. expressed as NSD (DAMMIF: NSD = 0.497 ± 0.027; DAMMIN: NSD = 0.519 ± 0.011; GAS-BORP: NSD = 0.918 ± 0.025). To rule out any effect of merging low and high concentration data, we also reconstructed an *ab initio* model for a single low concentration dataset (0.91 mg ml$^{-1}$) using DAMMIF, which basically yields the same overall shape as the models generated using merged data (Supplementary Fig. 11d). Theoretical scattering curves of monomeric *Te*LPOR apoprotein homology models (built as described in the Supplementary Discussion Section 1.4) were fitted to the experimental SAXS data using the program CRYSOL[52].

*Ab initio* models of the dimeric holoprotein were built using GASBORMX[49], which generates *ab initio* bead models of the symmetric oligomer while assuming a polydisperse sample containing a certain fraction of monomers. A fixed monomer fraction of 0.58 was used and 20 *ab initio* dimer models (under P2 symmetry) were built. The individual *ab initio* models were averaged and filtered using the DAMAVER programm[53]. All 20 models were grossly similar with an NSD value of 1.197 ± 0.025. For the corresponding averaged and filtered models of the apo and holoprotein envelope functions were determined using the program pdb2vol of the SITUS package[54]. The molecular mass was estimated from excluded volume of the filtered model by dividing the respective values by 2[28]. Atomic models of the dimeric ternary complex (built as described in the Supplementary Information) were evaluated by comparing the OLIGOMER-derived[55] theoretical scattering curve of monomer/dimer mixture to the experimental scattering data of the holoprotein. Form factor files for the monomer and the corresponding dimer were generated from the respective pdb file using FFMAKER. During homology model evaluation, missing N- and C-terminal structural elements, as well as internal, potentially disordered, elements, were modeled as flexible ensemble with EOM[56], using an initial pool of 10.000 random coil-like conformations, and by employing three rounds of genetic algorithm selection.

**TeLPOR activity assays**. Purified, reconstituted holoprotein samples were diluted with reaction buffer (20 mM Tris pH 7.5 buffer supplemented with 500 mM NaCl, 20% (v/v) glycerol) in 10 × 10 mm quartz-glass cuvettes. Additives (70 µM DTT, 0.03% (v/v) Triton X-100) were added separately to the reaction. Subsequently, the

assay mixture was equilibrated for 5 min at 25 °C in the dark. A blue-light emitting LED (450 nm; 2.6 mW cm$^{-2}$) was mounted on top of the cuvette, and light-dependent Pchlide turnover was initiated by illuminating the assay mixture employing cycles of 6 s blue-light illumination followed by 12 s in the dark during which a absorption spectrum (620 nm to 720 nm) was recorded. Pulsed illumination was achieved by a microcontroller-controlled LED driver (Arduino UNO (Smart Projects, Italy))[40]. Activity tests for the truncated *Te*LPOR variants, using either purified protein or cell-free lysates, were performed as described previously[40]. In brief, protein samples (*Te*LPOR wild type: 0.17 µM; truncated *Te*LPOR variants: 1.4 µM) or cell free lysates were diluted with reaction buffer, and NADPH (dissolved in reaction buffer) and Pchlide (dissolved in methanol) were added to a final concentration of 160 µM and 3.5 µM, respectively. Additives (70 µM DTT, 0.03% (v/v) Triton X-100) were added separately to the reaction. After 5 min incubation at 25 °C, samples were illuminated for variable times. Data was analysed using a home-written shell script, which filters and removes spectra that contain illumination events.

**Molecular modelling and molecular dynamics (MD) simulation**. The initial *Te*LPOR Rossmann-fold core domain model (residues 1-280) was built by using the previously published model of the LPOR of *Synechocystis* sp. (*Ss*LPOR)[14] as template, resulting in the *Te*LPOR Rossmann-fold core domain model (*Te*LPOR core). This model was subsequently extended C-terminally by loop modeling[57] with YASARA Structure Version 16.6.24[58,59], yielding the models *Te*LPOR-C$_{helix}$, *Te*LPOR-C$_{loop1helix}$ and *Te*LPOR-C$_{loop2helix}$ (Fig. 3a, Supplementary Fig. 10a). The C-terminal helical extension was manually built into the SAXS-derived DAMMIN/DAMMIF/GASBORP envelopes using YASARA Structure by relying on secondary structure information provided by secondary structure prediction (NPS@ consensus secondary structure prediction web server[60]; Supplementary Fig. 7a), the Phyre2 (Supplementary Fig. 7b)[61] and I-TASSER[62] homology modeling servers. Additionally, homology models of the *Te*LPOR full-length protein 1-322 (UniProt ID: Q8DLC1) were generated using the Phyre[2,61] and I-TASSER[62] web server. Both methods predict a NADPH binding Rossmann-fold core domain (residues 1-280) with a predominantly helical C-terminal extension (see Phyre[2] secondary structure prediction report; Supplementary Fig. 7b). This secondary structure assignment is also supported by consensus secondary structure prediction using the NPS@ web server[60] (Supplementary Fig. 7a). Phyre[2] identified 3 best homologs with 100% confidence and ~21% sequence identity (template X-ray structures: salutaridine reductase (PDB ID: 3O26)[63]; 20β-hydroxysteroid dehydrogenase (PDB ID: 1N5D)[31]; human CBR1 (PDB ID: 1WMA)[64] for the core domain. The C-terminal domain could be modeled as helical extension using the crystal structure of human mitochondrial 2,4-dienoyl-CoA reductase (PDB ID: 1W6U)[65] and human 17β-hydroxysteroid dehydrogenase (PDB ID: 1ZBQ) (unpublished) as template. I-TASSER predicted the helical C-terminal domain packed to the Rossmann-fold core domain (Supplementary Fig. 8).

Molecular dynamics simulations were performed employing the AMBER14 force field[66] for protein atoms and the general AMBER force field (GAFF)[67] for Pchlide and NADPH. The partial charges were derived using AM1-BCC point charge model[68]. To correct the covalent geometry, the structure was energy-minimized with using a 8.0 A force cutoff and the Particle Mesh Ewald algorithm[69] to treat long-range electrostatic interactions. After removal of conformational stress by a short steepest descent minimization, the procedure continued by simulated annealing (timestep 2 fs, atom velocities scaled down by 0.9 every 10th step) until convergence was reached, i.e. the energy improved by less than 0.05 kJ/mol per atom during 200 steps. The protein models were further relaxed by molecular dynamics simulations using a TIP3P[70] water box of 15 Å around the protein using pH 7.5 buffer supplemented with 500 mM NaCl and 20% (v/v) glycerol, according to the experimental conditions. The system was slowly heated to 298 K using a scaled Berendsen thermostat[71] and a pressure of 1 bar was applied, using the classical Berendsen barostat formula as implemented in YASARA during the equilibration and production phase. All angles and bond lengths to hydrogen were constraint[72] to enable faster simulations using a timestep of 5 fs and trajectory snapshots were saved ever 250 ps. The geometry of the trajectories were evaluated using YASARA Structure and VMD 1.9.2[73]. Figures were generated with Pymol 1.7.0.0 (Schrödinger, LCC, New York, NY, USA) or VMD 1.9.2.

**Sequence conservation analysis**. A sequence logo (Fig. 4b), to analyse sequence conservation within the LPOR enzyme family, was generated with Weblogo 3[74] employing a multiple sequence alignment of 315 LPOR sequences from plants and cyanobacteria. Residues are colored according to type, with hydrophobic residues shown in black, negatively charged residues in red, positively charged residues in blue, neutral amidic residues in purple and polar residues in green.

**Statistics and Reproducibility**. SAXS experiments were performed for three (apoprotein) or one (holoprotein) *Te*LPOR samples in a concentration series, showing low sample to sample variability. The complete datasets and the derived data are shown in Supplementary Figs. 4–6 and Supplementary Table 2. MWA-AUC analyses were performed for independently prepared apo- and holoprotein *Te*LPOR samples, yielding results in excellent agreement with similarly prepared SAXS samples. *Te*LPOR activity measurements were performed before all SAXS/

MWA-AUC samples to verify enzyme functionality. Truncated *Te*LPOR constructs were measured at least twice, for independently prepared crude extract samples and the respective purified proteins.

**Reporting Summary**. Further information on research design is available in the Nature Research Reporting Summary linked to this article.

## Data Availability

All source data used to generate the figures of this manuscript are available as Supplementary Materials (Supplementary Data 1 and 2). MWA-AUC raw data, pdb coordinate files of all models and MD simulation trajectories were deposited at the Zenodo data repository[75] (www.zenodo.org) under https://doi.org/10.5281/zenodo.3375375. Datasets generated and/or analysed during the current study are also available from the corresponding author upon reasonable request.

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

## Acknowledgements

We acknowledge the European Synchrotron Radiation Facility (ESRF) for provision of synchrotron radiation facilities and thank Dr. Martha Brennich for assistance in using beamline BM29. F.K. gratefully acknowledges Tina Franke (Nanolytics Instruments) and Doreen Freund (Nanolytics) for excellent technical assistance. JS and UK acknowledge funding by the Deutsche Forschungsgemeinschaft (DFG) (Grant "Origin, phylogeny, evolution and structural basis of light-driven protochlorophyllide reduction"; KR 3756/1-1). Simulations were performed with computing resources granted by JARA-HPC from RWTH Aachen University under project JARA0065.

## Author contributions

J.S. generated all constructs, expressed and purified the protein, isolated the Pchlide substrate, prepared samples for MWA-AUC and SAXS analyses, and performed activity measurements. F.K. performed MWA-AUC experiments, analysed and interpreted the corresponding data. M.B. and M.D.D. performed the protein modelling studies and molecular dynamics simulations and analysed the data. U.K. conceived the study, designed the experiments, performed SAXS measurements and analysed and interpreted the data with the help of A.M.S. K.E.J. and U.S. contributed to the manuscript. J.K., U.K. wrote the manuscript with contributions from all authors.

## Additional information

**Competing interests:** The authors declare no competing interests.

