## [Peer Review File · Communications Biology]

Reviewers' comments:

Reviewer #1 (Remarks to the Author):

COMMSBIO-19-0441-T

Protochlorophyllide (Pchl_{id}) reduction is the penultimate step of chlorophyll (Chl) a biosynthetic pathway. There are two distinct enzymes to catalyze Pchl_{id} reduction, one is the light-dependent Pchl_{id} reductase (LPOR) and the other is dark-operative Pchl_{id} oxidoreductase (DPOR). Because angiosperms employ LPOR as the sole Pchl_{id} reductase, seedlings of angiosperms grown in the dark become etiolated. While DPOR is a nitrogenase-like three-subunit enzyme that requires ferredoxin and ATP for catalysis, LPOR is a single subunit enzyme belonging to short-chain dehydrogenase (SDR) family and requires NADPH and light for catalysis. Crystal structures of DPOR have been reported in 2010, and the catalytic mechanism via radical intermediates and proton transfer has been also proposed in 2014 based on the structure. In contrast, there has yet no reports for crystal structure of LPOR so far while some simulated structures have been proposed based on the SDR structures. Nonetheless, detailed spectroscopic analyses using pump-probe and low-temperature techniques have revealed light-dependent hydride transfer followed by light-independent conformational changes. With the research progress in elucidating such reaction elementary processes, many biochemists, biophysicists, and plant physiologists, have been long-awaited its 3D structure to understand structural basis of the elementary processes over the past two decades. This manuscript by Schneidewind et al. reporting the a 3D structure of LPOR for the first time can be highly evaluated even low resolution.

The authors purified a His-tag LPOR from the thermophilic cyanobacterium *Synechococcus elongatus* in an *E. coli* overexpression system and analyzed apo and holo forms of LPOR by combination of biochemical and biophysical techniques (multi-wavelength absorbance analytical ultra-centrifugation, SAXS, and molecular dynamics). They showed that 1) the apo form of LPOR is a monomer, 2) the holo form of LPOR (the ternary complex of LPOR, Pchl_{id} and NADPH) is mainly a dimer, 3) the C-terminal domain of LPOR constitutes an α -helix that is flexibly mobile in the monomer state, 4) upon binding of Pchl_{id} and NADPH, the C-terminal α -helix interacts with the other protomer surface to form a dimer to stabilize the C-terminal domain, and 5) dimer formation is essential for the LPOR activity. If the dimer formation is essential for Pchl_{id} reduction, it would be reminiscent of the Pchl_{id} reduction by DPOR, in which the Pchl_{id} molecule held by the N- and B-subunits interacts with the C-terminal α -helix of B'-subunit (in the other symmetric unit) in the NB protein, the catalytic hetero-tetramer structure; (N-B)₂.

Because this reviewer is not a biophysicist, I cannot comment on the details in physical measurements and molecular dynamics calculations, but I believe that this manuscript would be a great pioneer paper. I have some comments shown below:

1) This reviewer agrees with that the apo-form of LPOR is a monomer, which is clearly shown by MWA-AUC and gel filtration chromatography. In contrast, analysis of the holo-form of LPOR, the ternary complex with Pchl_{id} and NADPH, was pretty complicated. MWA-AUC indicated that the monomer form occupied 89% and only the remaining 11% is dimer. Furthermore, the monomer form consisted of an apo- (67%) and a holo- (22%) forms. In understanding by this reviewer, this complicated composition suggests that the holo-forms of LPOR are under equilibrium between a monomer and a dimer. If so, the monomer-to-dimer arrow in Figure 5 should be bi-directional rather than uni-directional. The arrow from the apo-form to the holo-form may be also bi-directional. In addition, this reviewer is also wondering whether the monomeric and dimeric holo-forms are the same in specific activity.

2) No activity of the three C-terminal truncated variants seems to be consistent with the author's

claim that dimerization via the C-terminal domains is pre-requisite for LPOR activity. However, the cause of loss of activity in the truncated LPOR variants can be also considered to be other than their inability to dimerize. For example, severe decrease in Pchl_{id} binding ability could be considered. In order to show that the C-terminal domain is directly involved in the activity through dimer formation, showing a dominant negative effect in an experiment in which the wild-type LPOR sample is mixed with truncated LPOR variants may provide a persuasive result.

3) To prepare the holo-form of LPOR, Pchl_{id} and NADPH were added to the apo-form. This reviewer wonders whether the dimer and the monomer show the same stoichiometry of Pchl_{id} and NADPH to the LPOR protein. Data on the ratios of Pchl_{id} and NADPH to the LPOR protein may be important.

4) p. 14, line 387; Figure 4g rather than Figure 3g

Reviewer #2 (Remarks to the Author):

Overall, the manuscript by Schneidewind et al provides a cohesive set of experiments to derive structural information albeit at low resolution about the light-dependent protochlorophyllide oxidoreductase for which no medium- to high-resolution data is currently available.

While many of the individual interpretations seem open to debate, the combination of techniques and matching observations across techniques provides a compelling picture of the substrate-induced dimerization and activation of the enzyme.

Stylistically, the manuscript is heavy in technical jargon, making it quite a difficult read. The authors could consider simplifying the text in places particularly by reverting the excessive use of nouns in favour of more active verbs. Also suitably chosen abbreviations for the various constructs or complexes could prevent the repetition of long and awkward defining phrases.

Similarly, the long descriptive clauses in referring to the supplementary information is quite distracting. Would it not suffice to simply refer to a section or an image in the supplementary information?

One issue that I missed till quite late in the manuscript is the concept of the C-terminal helix stabilizing a domain-swapped LPOR dimer. This is a very plausible explanation for the stable LPOR dimer and should perhaps be emphasized earlier – in particular in figure 4, where this is so small that it only becomes obvious after zooming in to image. Perhaps using more contrasting colours for the dimer in Figure 4f – or indicating the domain-swapped helix in green – might emphasize the point more clearly.

Two of the C-terminal truncation variants of LPOR are a little problematic as they truncate the protein within the three-dimensional domain. This must inevitably result in production and solubility problems. Would it not have been more informative to include termination sites after A302 as this may have affected the stability of the LPOR dimer without affecting the solubility and potentially only marginally affecting the activity of the enzyme?

Overall, though, I believe this to be a valuable contribution to the scientific literature in this particular field and have no objections to its publication – once the comments above have been appropriately considered.

Wolf-Dieter Schubert

Reviewer #3 (Remarks to the Author):

The manuscript describes a low resolution structural analysis of a key light-driven enzyme in chlorophyll biosynthesis, protochlorophyllide oxidoreductase (POR). Due to its important biological role the POR enzyme has been studied in detail for many years but a full molecular understanding has not been possible, due to the lack of any structural information. Hence, the present paper is a timely addition and provides useful insights into POR structure. The authors clearly demonstrate that the protein forms larger multimeric structures upon binding of substrate and aimed to study the dimeric form by AUC and SAXS. Although the data is of good quality I have some major issues with the interpretation:

1. The authors state that the SAXS data is best fitted by a flexible C-terminal helix in the protein to give an 'extended' structure. However, it is well known in POR that it has a number of flexible regions, including for example an additional 25-30 residue region not found in similar enzymes. How can the authors exclude that these other regions could give rise to this flexible arm that is needed to fit the SAXS data.
2. Indeed, although I'm not a SAXS expert it appears that some of the models used to describe the SAXS data don't fit particularly well. For example the C-helix model in figs 3 h and i and figs 4 d and e don't fit particularly well to the data. Again is it not possible that other flexible regions of the protein could provide the the extended structures required to model the SAXS data.
3. I have major reservations about the C-terminal truncations that the authors have used to validate their model. These are fairly major truncations of the protein and hence, it is no surprise that the resulting enzymes are inactive. There could be many reasons why the truncated proteins are inactive (unfolded protein, changes in overall conformation of the monomer, altered binding sites, etc). Have the authors measured any CD spectra to check that the proteins are still folded? Does the enzyme still bind protochlorophyllide? In order to claim that this is evidence that the C-terminal region is involved in dimerisation the authors would at least have to repeat their SAXS or AUC measurements on these variants. Hence, currently I don't think these variants add anything to the overall model.

In conclusion, I think the work nicely shows dimerisation / oligomerisation upon binding and that this is driven by an extended conformation of the POR protein. However, I'm not convinced that this is necessarily caused by a flexible C-terminal arm in the protein as other models could also be used to explain this data.

Reviewers' comments:

Reviewer #1 (Remarks to the Author): COMMSBIO-19-0441-T

Protochlorophyllide (Pchl_{id}e) reduction is the penultimate step of chlorophyll (Chl) a biosynthetic pathway. There are two distinct enzymes to catalyze Pchl_{id}e reduction, one is the light-dependent Pchl_{id}e reductase (LPOR) and the other is dark-operative Pchl_{id}e oxidoreductase (DPOR). Because angiosperms employ LPOR as the sole Pchl_{id}e reductase, seedlings of angiosperms grown in the dark become etiolated. While DPOR is a nitrogenase-like three-subunit enzyme that requires ferredoxin and ATP for catalysis, LPOR is a single subunit enzyme belonging to short-chain dehydrogenase (SDR) family and requires NADPH and light for catalysis. Crystal structures of DPOR have been reported in 2010, and the catalytic mechanism via radical intermediates and proton transfer has been also proposed in 2014 based on the structure. In contrast, there has yet no reports for crystal structure of LPOR so far while some simulated structures have been proposed based on the SDR structures. Nonetheless, detailed spectroscopic analyses using pump-probe and low-temperature techniques have revealed light-dependent hydride transfer followed by light-independent conformational changes. With the research progress in elucidating such reaction elementary processes, many biochemists, biophysicists, and plant physiologists, have been long-awaited its 3D structure to understand structural basis of the elementary processes over the past two decades.

This manuscript by Schneidewind et al. reporting the a 3D structure of LPOR for the first time can be highly evaluated even low resolution. The authors purified a His-tag LPOR from the thermophilic cyanobacterium *Synechococcus elongatus* in an *E. coli* overexpression system and analyzed apo and holo forms of LPOR by combination of biochemical and biophysical techniques (multi-wavelength absorbance analytical ultra-centrifugation, SAXS, and molecular dynamics). They showed that 1) the apo form of LPOR is a monomer, 2) the holo form of LPOR (the ternary complex of LPOR, Pchl_{id}e and NADPH) is mainly a dimer, 3) the C-terminal domain of LPOR constitutes an α -helix that is flexibly mobile in the monomer state, 4) upon binding of Pchl_{id}e and NADPH, the C-terminal α -helix interacts with the other protomer surface to form a dimer to stabilize the C-terminal domain, and 5) dimer formation is essential for the LPOR activity.

1.1) If the dimer formation is essential for Pchl_{id}e reduction, it would be reminiscent of the Pchl_{id}e reduction by DPOR, in which the Pchl_{id}e molecule held by the N- and B-subunits interacts with the C-terminal α -helix of B'-subunit (in the other symmetric unit) in the NB protein, the catalytic hetero-tetramer structure; (N-B)2.

Authors response: *We thank the reviewer for this comment. The domain-swapped subunit arrangement suggested here for LPOR seems indeed reminiscent of the subunit arrangement observed previously in the DPOR structure, which is an interesting observation, given the structural dissimilarity of the two systems. The corresponding information has been included in the revised version of the manuscript (see page 17, lines 472-477).*

Because this reviewer is not a biophysicist, I cannot comment on the details in physical measurements and molecular dynamics calculations, but I believe that this manuscript would be a great pioneer paper. I have some comments shown below:

1.2) This reviewer agrees with that the apo-form of LPOR is a monomer, which is clearly shown by MWA-AUC and gel filtration chromatography. In contrast, analysis of the holo-form of LPOR, the ternary complex with Pchl_a and NADPH, was pretty complicated. MWA-AUC indicated that the monomer form occupied 89% and only the remaining 11% is dimer. Furthermore, the monomer form consisted of an apo- (67%) and a holo- (22%) forms. In understanding by this reviewer, this complicated composition suggests that the holo-forms of LPOR are under equilibrium between a monomer and a dimer. If so, the monomer-to-dimer arrow in Figure 5 should be bi-directional rather than uni-directional. The arrow from the apo-form to the holo-form may be also bi-directional. In addition, this reviewer is also wondering whether the monomeric and dimeric holo-forms are the same in specific activity.

Authors response: *As suggested, to illustrate the equilibrium, bi-directional arrows have been added for both reaction pathways in Figure 5 of the revised manuscript (see Figure 5, page 19). Unfortunately, all our efforts, using different buffers and SEC columns (either using FPLC or HPLC-SEC) to separate and purify the monomeric and dimeric holoprotein species failed. Apparently, the pigment is lost during purification, either due to dilution on the column and/or interaction with the column material. We therefore are not able to directly determine the specific activity of the monomeric and the dimeric holoprotein. To nevertheless address this issue, we now present additional MWA-AUC data in the revised version of the manuscript (see main manuscript page 7/8, lines 187-208; Supporting Results section 1.2; Figure S3g and h; page 18, lines 509-512). Here, the same holoprotein sample which was analyzed by MWA-AUC (containing a mixture of monomeric and dimeric holoprotein) was illuminated and subsequently analyzed by MWA-AUC. Those analyses revealed that after illumination both the monomeric as well as the dimeric species co-sediment with small amounts of the Chlide reaction product, likely in form of Chlide/TeLPOR binary complexes (Heyes and Hunter. 2004. Biochemistry. 43: 8265-8271). Moreover, we observed that illumination alters the monomer:dimer distribution in favor of the monomer (from 2:1 in the dark to 3-4.5:1 after illumination). This observation independently supports our model presented in Figure 5 of the main manuscript, corroborating previous studies on LPORs (Kahn, Boardman, Thorne. 1970. J. Mol. Biol. 48:85-101; Reinbothe, Pollmann, Reinbothe. 2003. J. Biol. Chem. 278:800-806), thus providing a rational for light-dependent disintegration of prolamellar bodies.*

In conclusion, these results suggest that either both the holoprotein monomer as well as the holoprotein dimer show light-dependent activity, or that the monomeric Chlide/TeLPOR binary complex, detected by MWA-AUC, results from the dissociation of the product-containing dimer. Given the altered monomer:dimer distribution which we observed for the illuminated holoprotein sample (monomer:dimer 3-4.5:1; Figure S3g) as compared to the dark-reconstituted holoprotein (monomer:dimer 2:1; Figure 1d, Figure S3c), we would favor the latter interpretation. However, since we were not able to separate the monomeric and dimeric holoprotein species chromatographically, to independently test their activity, we are unable to clearly delineate between the two possibilities.

The outlined results are discussed in detail in the revised SI (Supporting Results section 1.2; Figure S3g and h) and are briefly discussed in the main manuscript (see page 7/8, lines 187-208; page 18, lines 509-512). To include the possibility that both the holoprotein monomer

and dimer could be active, we further modified Figure 5 to include this possibility (discussed briefly now also in the revised manuscript page 18, lines 509-512; page 19, lines 522-524).

1.3) No activity of the three C-terminal truncated variants seems to be consistent with the author's claim that dimerization via the C-terminal domains is pre-requisite for LPOR activity. However, the cause of loss of activity in the truncated LPOR variants can be also considered to be other than their inability to dimerize. For example, severe decrease in Pchlide binding ability could be considered. In order to show that the C-terminal domain is directly involved in the activity through dimer formation, showing a dominant negative effect in an experiment in which the wild-type LPOR sample is mixed with truncated LPOR variants may provide a persuasive result.

Authors response: *Based on the data that we have, which is also presented and discussed in the originally submitted manuscript (Figure 4h, Figure S21), it is in our opinion clear that C-terminal truncation of TeLPOR "either influences proper folding of the protein [which includes impairment of Pchlide binding] or impairs the formation of the dimeric photoactive TeLPOR/NADPH/Pchlide ternary holoprotein complex". This was already briefly stated in the originally submitted manuscript (see page 14/15, line 396/397 of the original submission). We agree with the reviewer's assessment that the present data does not allow unequivocal distinction between the different effects, and regret that we have not made this more clear earlier. Moreover, as shown in the originally submitted manuscript, all three truncated variants express very badly and cannot be purified to homogeneity (Figure S20). At best, TeLPOR-Δ21 (Figure S20b) can be obtained at about 50% purity. This renders the suggested experiment unlikely to yield a meaningful outcome, because the samples that could be tested are not homogeneous. In consultation with Dr. Brooke LaFlamme, chief editor at Communications Biology, we therefore did not perform further experiments but instead moderated our statements regarding the truncation data (see abstract: half-sentence about the truncation data was removed, see page 2; end of introduction: the statement about the truncation data was moderated, see page 4, lines 102-103; Results section: the whole paragraph about the truncation data, including its heading, was moderated, see page 15-16, lines 402-420; Discussion section: sentence moderated accordingly, page 18, lines 493-496)*

1.4) To prepare the holo-form of LPOR, Pchlide and NADPH were added to the apo-form. This reviewer wonders whether the dimer and the monomer show the same stoichiometry of Pchlide and NADPH to the LPOR protein. Data on the ratios of Pchlide and NADPH to the LPOR protein may be important.

Authors response: *The only way to address this question is to compare the MWA extracted spectra. Figure S3f depicts the spectra extracted from the MWA-AUC run of the holoprotein monomer species (red line) and the holoprotein dimer species (green line). The composition of holoprotein monomer/dimer are approximately identical as deduced from similar spectral properties. i.e. a comparable ratio of absorbance at 330 nm and 440 nm for both species (approximately 1.5-1.6, Figure S3f).*

1.5) p. 14, line 387; Figure 4g rather than Figure 3g

Authors response: *We thank the reviewer for spotting this typo. The mistake has been corrected in the revised version of the manuscript (see page 15, line 406).*

Reviewer #2 (Remarks to the Author):

Overall, the manuscript by Schneidewind et al provides a cohesive set of experiments to derive structural information albeit at low resolution about the light-dependent protochlorophyllide oxidoreductase for which no medium- to high-resolution data is currently available.

While many of the individual interpretations seem open to debate, the combination of techniques and matching observations across techniques provides a compelling picture of the substrate-induced dimerization and activation of the enzyme.

2.1) Stylistically, the manuscript is heavy in technical jargon, making it quite a difficult read. The authors could consider simplifying the text in places particularly by reverting the excessive use of nouns in favour of more active verbs. Also suitably chosen abbreviations for the various constructs or complexes could prevent the repetition of long and awkward defining phrases.

***Authors response:** For better readability certain sentences, in particular in the Results section were shortened as suggested (see page 4/5 lines 111-112; page 7, lines 181-183; page 8, lines 211-213; page 8, lines 215-217; page 11, page 11, lines 283-285; page 13, line 339-341, page 15, lines 389). However, we decided against choosing abbreviations for the different constructs. In fact, apart from the truncated variants that are only briefly mentioned in one paragraph, our study deals with the same protein construct in two forms (apo- and holo). We believe that while abbreviations might shorten the text, their use would not contribute to better readability or easier understanding. Instead we carefully revised the manuscript to avoid the unnecessary repetition of defining phrases after their first definition (e.g. using apoprotein instead of pigment-free TeLPOR apoprotein and holoprotein instead of TeLPOR/NADPH/Pchlde ternary holoprotein complex) (see page 3, line 50 and 58/59; page 4, line 91, 95, 108 and 109; page 5, line 112; page 5, line 121, 123, 130 and 132; page 6, line 148 and 165, page 8, line 211, 215, 223; page 13, line 329; page 17, line 453; page 18, line 50; page 19, line 515 and 519; page 25, line 702 and 719).*

2.2) Similarly, the long descriptive clauses in referring to the supplementary information is quite distracting. Would it not suffice to simply refer to a section or an image in the supplementary information?

***Authors response:** Since the SI contains additional description/discussion of results, necessary to understand our conclusions, we referenced the corresponding subsections of the SI Supporting Results by subheadings. We agree that this does make certain sentences difficult to read. To make cross-referencing easier and the text shorter, we now introduced section numbers in the Supporting Results section of the SI and now cross-reference only by section number. We hope that this enables easier reading (see page 5, line 119; page 7, line 183; page 8, line 208 and 217; page 9, line 237; page 10, line 261,270/217 and 272; page 13, line 338 and 346).*

2.3) One issue that I missed till quite late in the manuscript is the concept of the C-terminal helix stabilizing a domain-swapped LPOR dimer. This is a very plausible explanation for the stable LPOR dimer and should perhaps be emphasized earlier – in particular in figure 4,

where this is so small that it only becomes obvious after zooming in to image. Perhaps using more contrasting colours for the dimer in Figure 4f – or indicating the domain-swapped helix in green – might emphasize the point more clearly.

Authors response: *We thank the reviewer for this comment. We now include this information earlier in the revised manuscript (see page 15, lines 397-399) and comment on the issue also in the discussion section (see page 17, lines 458,459 and 472-477). Figure 4f has been enlarged for better visibility and the C-terminal helix is now colored in green (see Figure 4f, page 14).*

2.4) Two of the C-terminal truncation variants of LPOR are a little problematic as they truncate the protein within the three-dimensional domain. This must inevitably result in production and solubility problems. Would it not have been more informative to include termination sites after A302 as this may have affected the stability of the LPOR dimer without affecting the solubility and potentially only marginally affecting the activity of the enzyme?

Authors response: *Our truncation experiments aimed at showing the role of the C-terminal segment, not covered by existing homology models. Therefore, we inserted terminations sites equally spaced within the whole protein segment. Based on the data that we have, which is also presented and discussed in the manuscript (Figure 4h, Figure S21), it is in our opinion clear that C-terminal truncation of TeLPOR “either influences proper folding of the protein [which includes impairment of Pchl_{ide} binding] or impairs the formation of the dimeric photoactive TeLPOR/NADPH/Pchl_{ide} ternary holoprotein complex” (see page 14/15, line 396/397 of the original submission). We agree with the reviewer’s assessment that the present data does not allow unequivocal distinction between the different effects. This we make more clear now in the revised version of the manuscript (see page 16, lines 417-420). In consultation with Dr. Brooke LaFlamme, chief editor at Communications Biology, we did not generate additional variants, but instead moderated our statements regarding the truncation data (see abstract: half-sentence about the truncation data was removed, see page 2; end of introduction: the statement about the truncation data was moderated, see page 4, lines 102-103; Results section: the whole paragraph about the truncation data, including its heading, was moderated, see page 15-16, lines 402-420; Discussion section: sentence moderated accordingly, page 18, lines 493-496).*

Overall, though, I believe this to be a valuable contribution to the scientific literature in this particular field and have no objections to its publication – once the comments above have been appropriately considered.

Wolf-Dieter Schubert
University of Pretoria, Pretoria, South Africa

Reviewer #3 (Remarks to the Author):

The manuscript describes a low resolution structural analysis of a key light-driven enzyme in chlorophyll biosynthesis, protochlorophyllide oxidoreductase (POR). Due to its important biological role the POR enzyme has been studied in detail for many years but a full molecular understanding has not been possible, due to the lack of any structural information. Hence, the present paper is a timely addition and provides useful insights into POR structure. The authors clearly demonstrate that the protein forms larger multimeric structures upon binding of substrate and aimed to study the dimeric form by AUC and SAXS. Although the data is of good quality I have some major issues with the interpretation:

3.1) The authors state that the SAXS data is best fitted by a flexible C-terminal helix in the protein to give an 'extended' structure. However, it is well known in POR that it has a number of flexible regions, including for example an additional 25-30 residue region not found in similar enzymes. How can the authors exclude that these other regions could give rise to this flexible arm that is needed to fit the SAXS data.

***Authors response:** We believe that the reviewer primarily refers to a protein segment, which differentiates LPORs from other short-chain dehydrogenases (residues 154 to 185 of TeLPOR). To the best of our knowledge, this was initially pointed out by Townley and coworkers in 2001 (Townley et al. 2001. *Proteins*. 44:329-335). It is true that this segment, which we will call LPOR insertion loop in the following, is not covered by the original template used by Townley et al for homology modelling. The authors, modeled this segment partially as α -helical. In our model this corresponds to the residues 172-182, which we therefore also modeled as α -helix. This is in excellent agreement with recent NMR data provided in the doctoral thesis of David Robert Armstrong (2014, University of Sheffield, UK)(see page 150, Figure 7.14B), where TALOS-N based secondary structure predictions suggested α -helical secondary structure for the residues around residue 170. To address the issue raised by this reviewer, we performed a number of additional analyses.*

First of all, in the already presented MD simulations, the corresponding region does on average show the same mobility (RMSF: $0.3 \pm 0.1 \text{ \AA}$) as the Rossmann-fold core domain excluding the immediate termini (residues 9 to 285; $0.2 \pm 0.1 \text{ \AA}$)(Figure 4c). The comparison of the apoprotein monomer and holoprotein dimer simulations reveals only marginally increased mobility in the apo-protein monomer simulations as compared to the dimer for the corresponding region (Figure 4c). Moreover, the α -helix segment (residues 172-182) of the insertion loop retains α -helical secondary structure in our MD simulations (Figure S10, c-d; marked now by black boxes). This suggests that at least the helical part of the LPOR insertion loop is structurally stable during simulations. The corresponding information has been added to the revised version of the Supporting Information (see Supporting Results section 1.4.4).

*To evaluate if a disordered insertion loop could account for our SAXS data, we performed additional calculations with EOM, which allows the modeling of certain segments of a structure as flexible, i.e. as a disordered ensemble, and compares this ensemble data against experimental SAXS data. First, we performed disorder predictions for TeLPOR using the MetaDisorder web service, which combines multiple disorder prediction algorithms (Kozlowski et al. 2012. *BMC Bioinformatics* 13: 111). Here, potentially disordered regions were identified at the immediate N-terminus, for the C-terminal extension (residue 285*

onwards) and, albeit with weak disorder tendency, also for the potential LPOR insertion loop (residues 160 to 185) (Figure S9c). Subsequently, we evaluated different scenarios, which differed only with respect to the regions that were modeled as disordered ensemble:

In scenario 1 (model 1) the complete insertion loop (residues 154-185; defined by Townley and coworkers (Townley et al. 2001. *Proteins*. 44:329-335)(Figure S9e, highlighted in red) and the C-terminal extension were allowed to be modeled as disordered ensemble (Figure S9e, highlighted in blue), while in scenario 2 (model 2), the C-terminal extension was kept rigid as in our best C-terminally extended model, whereas the insertion loop was modeled as disordered ensemble (Figure S9e). In scenario 3 (model 3) only a part of the insertion loop (residues 150-160) (Figure S9f, highlighted in red), as well as the complete C-terminal extension (Figure S9d, highlighted in blue) was assumed to be disordered. In scenario 4 (model 4) the short insertion loop segment (Figure S9f, highlighted in red) was assumed disordered, while the C-terminal extension was kept rigid. The latter two scenarios are based on unpublished NMR data presented as part of the doctoral thesis of David Robert Armstrong (2014, University of Sheffield, UK), which by NMR relaxation experiments showed that the residues between 150 and 160 show increased mobility. All four scenarios were scored against our experimental SAXS data (Figure S9g,h) yielding χ values between $\chi = 2.124$ (model 1) and $\chi = 2.620$ (model 4), which are significantly worse as compared to our best initial model in which both the C-terminal extension as well as the insertion loop were kept rigid ($\chi = 1.654$; see Figure S9g,h; +C_{helix} model shown for comparison).

In conclusion, modeling the insertion loop segment as disordered ensemble did not yield a improved model, i.e. compared to our best TeLPOR+C_{helix} model. This essentially suggests that a flexible, disordered insertion loop is unlikely to account for our SAXS data. Given the low resolution nature of SAXS data, we, however, cannot rule out completely that other structural models exist, in which a disordered insertion loop could account for our SAXS data. To give credit to this possibility we now include a corresponding statement at the end of the discussion section (page 19, lines 526-531) of the revised manuscript. Nevertheless, we strongly believe, that we now covered and modeled all potential alternative options, and that the final model possessing a protruding α -helical extension best describes our data.

All of the just mentioned additional analyses are now described and discussed in the manuscript (see page 10, line 261-271 and page 11, line 277-279 and the SI (see Supporting Results section 1.4.4). The corresponding data is shown in Figure S9c-h in the revised SI.

3.2) Indeed, although I'm not a SAXS expert it appears that some of the models used to describe the SAXS data don't fit particularly well. For example the C-helix model in figs 3 h and i and figs 4 d and e don't fit particularly well to the data. Again is it not possible that other flexible regions of the protein could provide the the extended structures required to model the SAXS data.

Authors response: SAXS is particularly sensitive to determine protein shape at low resolution. Information about protein shape is visible primarily in the small angle range up to around $q = 0.2 \text{ \AA}^{-1}$ for our case here (Figure 3h and i for the monomer and in Figure 4d and e for the dimer). Our models describe that low q -range well, which is illustrated by the obtained χ -values that are between 1 and 2 for the best structural models. Higher scattering vectors inform about atomistic orientations and illustrate that our models slightly differ at atomistic

scale. However, errors at the high q -range above 0.2 \AA^{-1} are rather large and theoretical curves agree with the experimental data within the uncertainty. Furthermore, our best structural models agree well with low-resolution shapes that have been obtained via *ab initio* modelling from the experimental SAXS data in a different approach (see Figure 3j for the monomer and Figure 4f for the dimer). See also our response to Reviewer 3 comment 3.1.

3.3) I have major reservations about the C-terminal truncations that the authors have used to validate their model. These are fairly major truncations of the protein and hence, it is no surprise that the resulting enzymes are inactive. There could be many reasons why the truncated proteins are inactive (unfolded protein, changes in overall conformation of the monomer, altered binding sites, etc). Have the authors measured any CD spectra to check that the proteins are still folded? Does the enzyme still bind protochlorophyllide? In order to claim that this is evidence that the C-terminal region is involved in dimerisation the authors would at least have to repeat their SAXS or AUC measurements on these variants. Hence, currently I don't think these variants add anything to the overall model.

Authors response: *As also outlined above and in our response to the editor decision; also shown in the originally submitted manuscript, all three truncated variants express very badly and cannot be purified to homogeneity (Figure S20). At best, TeLPOR- Δ 21 (Figure S20b) can be obtained at about 50% purity. This renders the suggested experiments (CD; SAXS, AUC) impossible to perform without yielding ambiguous information only, e.g. because the samples that could be tested are not homogeneous. Those experiments are thus unlikely to yield a meaningful outcome. As outlined above, in consultation with Dr. Brooke LaFlamme, chief editor at Communications Biology, we therefore have moderated our statements regarding the truncation data (see abstract: half-sentence about the truncation data was removed, see page 2; end of introduction: the statement about the truncation data was moderated, see page 4, lines 102-103; Results section: the whole paragraph about the truncation data, including its heading, was moderated, see page 15-16, lines 402-420; Discussion section: sentence moderated accordingly, page 18, lines 493-496).*

3.4) In conclusion, I think the work nicely shows dimerisation / oligomerisation upon binding and that this is driven by an extended conformation of the POR protein. However, I'm not convinced that this is necessarily caused by a flexible C-terminal arm in the protein as other models could also be used to explain this data.

Authors response: *See our response to Reviewer 3 comment 3.1. Given the low resolution nature of SAXS data, we can of course not rule out completely that other structural models exist, in which a disordered insertion loop could account for our SAXS data. However, we strongly believe, that we now covered and modeled all potential alternative options, and that the final model possessing a protruding α -helical extension best describes our data. This issue is now commented on at the end of the discussion section (see page 19, lines 526-531).*

References

Townley HE, Sessions RB, Clarke AR, Dafforn TR, Griffiths WT (2001) Protochlorophyllide oxidoreductase: A homology model examined by site-directed mutagenesis. *Proteins* 44: 329-335.

Armstrong DR (2014) Structural and Functional Studies of the Light-Dependent Protochlorophyllide Oxidoreductase Enzyme. doctoral thesis (The University of Sheffield).

Kozlowski LP, Bujnicki JM (2012) Metadisorder: A meta-server for the prediction of intrinsic disorder in proteins. *BMC Bioinformatics* 13: 111

Heyes DJ, Hunter CN (2004) Identification and characterization of the product release steps within the catalytic cycle of protochlorophyllide oxidoreductase. *Biochemistry* 43: 8265-8271

REVIEWERS' COMMENTS:

Reviewer #1 (Remarks to the Author):

This reviewer found that the authors made revisions properly according to the reviewer's comments and criticisms. Only one thing this reviewer would like to point out is that the panels g and h in Figure S3 should be placed on left and right columns, respectively, according to the arrangement of other panels in this Figure.

Reviewer #2 (Remarks to the Author):

Stylistically, the text is still excessively convoluted. To ensure maximum readability and hence impact, the authors should simplify and shorten the text to its bare bones.

To illustrate my point, I have roughly edited the first part of the manuscript (see attached file).

Other than that I have no issue with the experiments and the interpretation of the results.

Reviewer #3 (Remarks to the Author):

The authors have obviously taken great care to address all of the concerns raised by the reviewers but I still feel two main issues remain.

1. The issue I raised previously that other flexible regions of the protein could easily be used to explain the extended structure still exists. The authors have carried out further evaluation of the data by looking at the additional loop in POR that is not present in other SDR proteins (residues 154-185) and their finding that modeling of this insertion does not yield an improved model is sound. However, my point was that there are likely to be several of these disordered loops in POR that could potentially become ordered upon binding. Presumably this is the main reason why it has been so difficult to crystallise the enzyme. I just highlighted this insertion region as one example. It is known in the literature that many of the SDR family of proteins have disordered regions that become ordered upon substrate binding and that these act as a 'lid' to bind the substrate in the correct orientation for catalysis. Have the authors not been able to identify any such regions in POR from their homology modelling? I appreciate that it becomes impossible to analyse all possible disordered loops in the protein but is it not at least possible to identify potential loops that are involved in Pchlide binding from the homology model. The original homology model that the authors used didn't even have the C-terminal region present but still had Pchlide bound. How has this been rationalised?

2. As it stands the C-terminal truncation data don't really tell you anything apart from the fact that they yield inactive or insoluble protein. In my opinion, more conservative truncations (e.g. 5-10 residues) of the C-terminal region would have been much more informative. These are likely to have much less impact on the overall folding of the protein than the large truncations performed by the authors. Have the authors tried shorter truncations? It is possible that they will be easier to produce and hence it would then facilitate SAXS measurements which should provide conclusive proof of the C-terminal model.

Overall, the paper shows conclusively that an extended conformation of the POR protein is formed upon binding that is likely to be caused by a rigidification of a disordered region of the protein. This then leads to dimerisation / oligomerisation. However, I'm still inclined to be very cautious about overstating that this process is driven by a flexible C-terminal arm in the protein.

Reviewer #1 (Remarks to the Author):

1.1 This reviewer found that the authors made revisions properly according to the reviewer's comments and criticisms. Only one thing this reviewer would like to point out is that the panels g and h in Figure S3 should be placed on left and right columns, respectively, according to the arrangement of other panels in this Figure.

Authors response: Thank you for spotting this mistake! We have swapped the two panels as suggested (see Figure S3 in the revised SI).

Reviewer #2 (Remarks to the Author):

2.1 Stylistically, the text is still excessively convoluted. To ensure maximum readability and hence impact, the authors should simplify and shorten the text to its bare bones. To illustrate my point, I have roughly edited the first part of the manuscript (see attached file). Other than that I have no issue with the experiments and the interpretation of the results.

Authors response: We thank this reviewer for editing the first part of our manuscript for clarity. We have incorporated all suggestions and have carefully edited the remaining manuscript accordingly.

Reviewer #3 (Remarks to the Author):

The authors have obviously taken great care to address all of the concerns raised by the reviewers but I still feel two main issues remain.

3.1 The issue I raised previously that other flexible regions of the protein could easily be used to explain the extended structure still exists. The authors have carried out further evaluation of the data by looking at the additional loop in POR that is not present in other SDR proteins (residues 154-185) and their finding that modeling of this insertion does not yield an improved model is sound. However, my point was that there are likely to be several of these disordered loops in POR that could potentially become ordered upon binding. Presumably this is the main reason why it has been so difficult to crystallise the enzyme. I just highlighted this insertion region as one example. It is known in the literature that many of the SDR family of proteins have disordered regions that become ordered upon substrate binding and that these act as a 'lid' to bind the substrate in the correct orientation for catalysis.

Authors response: To acknowledge this possibility, which we cannot exclude explicitly due to limitations in our modelling/docking strategy, we have included the following short paragraph within the revised discussion section: “Secondly, flexible regions of the protein might become ordered upon ternary complex formation, which is a known feature of other SDR enzymes^{35,38}. This in turn might present a novel interface for dimerization, which is absent in the apoprotein, thereby driving dimerization. This model could not be evaluated as it assumes substantial conformational changes in the monomer due to Pchlide/NADPH binding, which we cannot assess as our starting TeLPOR monomer model already contained bound Pchlide and NADPH (see Supplementary Discussion section 1.4.1). Please note, however, that our disorder prediction for TeLPOR did not detect any potential disordered regions other than the LPOR insertion loop and the C-terminal extension. Likewise, dimerization might induce further larger scale conformational changes within the monomer, which are not accommodated by the ClusPro rigid-body docking methodology²⁹”

3.2 Have the authors not been able to identify any such regions in POR from their homology modelling? I appreciate that it becomes impossible to analyse all possible disordered loops in the protein but is it not at least possible to identify potential loops that are involved in Pchlide binding from the homology model.

Authors response: See also our response to comment 3.1. We were not able to identify other potentially disordered (flexible) loops involved in Pchlide binding. First, disorder predictions, which we performed during the previous revision (see Figure S9c) did not identify any other potentially disordered regions with sufficient confidence, except for the mentioned LPOR insertion loop (residues 154-185) and the C-terminal likely helical extension (see Figure 9c; disorder tendency >0.5 as predicted by the MetaDisorder webserver). Secondly, MD simulations of the apoprotein did not reveal large scale conformational changes, or regions with increased mobility, within the Rossmann-fold core domain (see Figure 3d to f and Figure S10). At last, the most pertinent argument for the absence of other disordered regions in the apoprotein, however, comes from the already mentioned doctoral thesis of David Robert Armstrong who performed NMR relaxation experiments on the TeLPOR apoprotein (unpublished data). Here only the residues 150-160 (LPOR insertion loop) and the section from 285 onwards (C-terminal extension) showed elevated T_2 relaxation rates, indicative of increased mobility. To make this more clear we have slightly modified the corresponding paragraph of the Supporting Results Section (see Section 1.4.4. It now reads: “Here,

potentially disordered regions were only identified at the immediate N-terminus, for the C-terminal extension (residue 285 onwards) and, albeit with weak disorder tendency, also for the potential LPOR insertion loop (residues 160 to 185) (Figure S9c)."

3.3 The original homology model that the authors used didn't even have the C-terminal region present but still had Pchlde bound. How has this been rationalised?

Authors response: As already mentioned in the Supporting Results Section 1.4.1, our homology model was built based on the Synechocystis sp. homology model presented by Townley and co-workers (Townley et al. 2001. Proteins. Structure Function and Bioinformatics. 44:329-335). This model already contained NADPH and Pchlde and thus represents the Rossmann-fold core domain of the holoprotein monomer. For SAXS data evaluation and MD simulations of the apoprotein monomer we removed Pchlde and NADPH from our model. To make this more clear we slightly modified the corresponding paragraph in the SI (see Section 1.4.1). It now reads: "This model, to which we refer to as TeLPOR core domain, represents the monomer subunit of the enzyme as ternary (holoprotein) complex with bound NADPH and Pchlde¹¹. The corresponding apoprotein monomer was generated by removing NADPH and Pchlde from the model"

3.4 As it stands the C-terminal truncation data don't really tell you anything apart from the fact that they yield inactive or insoluble protein. In my opinion, more conservative truncations (e.g. 5-10 residues) of the C-terminal region would have been much more informative. These are likely to have much less impact on the overall folding of the protein than the large truncations performed by the authors. Have the authors tried shorter truncations? It is possible that they will be easier to produce and hence it would then facilitate SAXS measurements which should provide conclusive proof of the C-terminal model.

Authors response: To address this issue we have further moderated our statements about the conclusions drawn from the truncations studies. We now state in the Results section: "This might indicate that the C-terminal protein segment is important for the proper folding of TeLPOR. However, further variants, i.e. with more conservative, shorter, truncations, would have to be studied to ascertain the role of the C-terminal extension for the function and dimerization of TeLPOR." Likewise the corresponding Discussion section now reads: "This notion is supported by our docking studies and the observation that removal of the C-terminal extension resulted in poorly expressed or inactive protein (Figure 5f, Supplementary Fig. 20b-d and 21). However, to ascertain the role of the C-terminal extension for TeLPOR function and dimerization further studies would be needed."

3.5 Overall, the paper shows conclusively that an extended conformation of the POR protein is formed upon binding that is likely to be caused by a rigidification of a disordered region of the protein. This then leads to dimerisation / oligomerisation. However, I'm still inclined to be very cautious about overstating that this process is driven by a flexible C-terminal arm in the protein.

Authors response: To further moderate our statement that indeed the model that we present is the only conceivable one we now also briefly discuss the alternative model suggested by this reviewer (see our response to comment 3.1). In addition we now conclude the Discussion section with the following statement: "Please note that this mechanistic proposal is solely based on the here presented consensus TeLPOR apoprotein monomer and holoprotein dimer

models as well as simulation data. Those models, selected from several likely alternatives, are best-fitting to our experimental SAXS data, they are physically feasible and remain stable in MD simulations. Given the low resolution nature of SAXS data, we, however, cannot not rule out that other structural models, not covered by our alternative modelling approaches, exist, that would equally well account for the presented SAXS data. High-resolution X-ray or NMR-structures will be needed to address this issue unequivocally. ”